# EAG3R: Event-Augmented 3D Geometry Estimation for Dynamic and Extreme-Lighting Scenes

**Xiaoshan Wu**[1,*], **Yifei Yu**[1,*,†], **Xiaoyang Lyu**[1], **Yihua Huang**[1],
**Bo Wang**[1], **Baoheng Zhang**[1], **Zhongrui Wang**[2,‡], **Xiaojuan Qi**[1,‡]

[1]The University of Hong Kong    [2]Southern University of Science and Technology

[*] Equal contribution, ordered alphabetically,    [†] Project lead,    [‡] Corresponding author

## Abstract

Robust 3D geometry estimation from videos is critical for applications such as autonomous navigation, SLAM, and 3D scene reconstruction. Recent methods like DUSt3R demonstrate that regressing dense pointmaps from image pairs enables accurate and efficient pose-free reconstruction. However, existing RGB-only approaches struggle under real-world conditions involving dynamic objects and extreme illumination, due to the inherent limitations of conventional cameras. In this paper, we propose **EAG3R**, a novel geometry estimation framework that augments pointmap-based reconstruction with asynchronous event streams. Built upon the MonST3R backbone, EAG3R introduces two key innovations: (1) a retinex-inspired image enhancement module and a lightweight event adapter with SNR-aware fusion mechanism that adaptively combines RGB and event features based on local reliability; and (2) a novel event-based photometric consistency loss that reinforces spatiotemporal coherence during global optimization. Our method enables robust geometry estimation in challenging dynamic low-light scenes without requiring retraining on night-time data. Extensive experiments demonstrate that EAG3R significantly outperforms state-of-the-art RGB-only baselines across monocular depth estimation, camera pose tracking, and dynamic reconstruction tasks.

## 1 Introduction

Estimating geometry from videos or images is a fundamental problem in 3D vision, with broad applications in camera pose estimation, novel view synthesis, geometry reconstruction, and 3D perception. These capabilities are crucial in downstream scenarios such as autonomous driving, SLAM, virtual environments, and robotic navigation. Recent methods like DUSt3R [64] have shown that regressing dense pointmaps from image pairs using transformer-based foundation models enables accurate and efficient pose-free 3D reconstruction. This paradigm has sparked a growing trend toward addressing various challenging scenarios, such as longer image sequences [59, 62, 60], dynamic scenes [72, 10, 27, 55], and integration with techniques like Gaussian Splatting [16, 52, 18].

However, in real-world applications such as autonomous driving in the wild, which often involve fast motion and rapidly changing illumination, RGB cameras—dependent on long exposure times for imaging—face significant challenges, including blur, out-of-focus artifacts, overexposure, and underexposure. Consequently, the resulting low-quality images hinder reliable geometry estimation.

Event cameras, on the other hand, provide asynchronous measurements of pixel-level brightness changes with high temporal resolution and dynamic range. They have demonstrated strong resilience in challenging conditions such as fast motion and extreme illumination [17, 28, 47]. Prior work has leveraged event streams in 3D tasks such as depth estimation [4, 79, 40], surface reconstruction [8, 9],

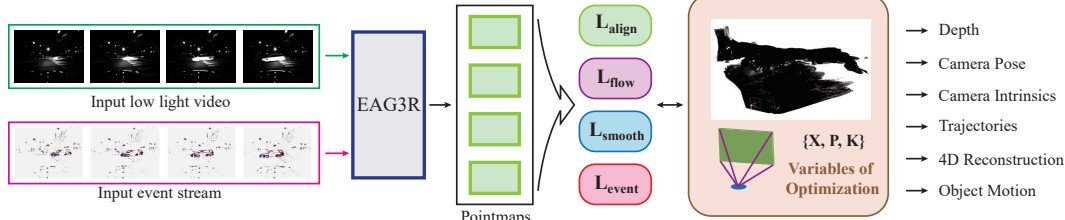

Figure 1: **EAG3R pipeline for event-augmented dynamic 3D reconstruction.** EAG3R processes a low-light video and its corresponding event stream within a temporal window, extracting pairwise pointmaps for each frame pair. These pointmaps are jointly optimized under alignment, flow, smoothness, and event-based consistency losses to recover a global dynamic point cloud and per-frame camera poses and intrinsics $\{X, P, K\}$. This unified representation enables efficient downstream tasks such as depth estimation and camera pose estimation, under challenging lighting conditions.

and neural rendering [48, 25], but their integration into modern learning-based geometry pipelines remains limited.

In this paper, we propose **EAG3R**, an event-augmented MonST3R framework to enhance pointmap-based 3D geometry estimation under dynamic and extremely low-light conditions. Built upon the MonST3R [72] backbone, EAG3R introduces two key innovations: (1) a lightweight event adapter with Signal-to-Noise Ratio (SNR)-aware fusion mechanism that adaptively integrates event and image features based on local reliability, and (2) an event-based photometric consistency loss that enforces alignment between predicted motion-induced brightness changes and event-observed brightness changes during global optimization. These components enable EAG3R to remain robust in scenarios where conventional RGB-only pipelines fail.

We evaluate EAG3R on the MVSEC dataset [78], conducting extensive comparisons on depth estimation, camera pose tracking, and dynamic reconstruction in extreme low-light conditions. Results show that EAG3R significantly outperforms existing baselines, including DUSt3R [64], MonST3R [72], and Easi3R [10] variants, even in a zero-shot nighttime setting.

**Our main contributions are as follows:**

- We propose **EAG3R**, the first event-augmented pointmap-based geometry estimation framework, which integrates asynchronous event streams with RGB-based reconstruction to handle dynamic scenes under extreme low-light conditions.

- We design a **plug-in event perception module** that integrates RGB and event data via: (1) a Retinex-based enhancer for visibility recovery and SNR map prediction; (2) a lightweight Swin-Transformer-based event adapter; and (3) an SNR-aware fusion scheme for adaptive feature integration.

- We develop a **novel event-based photometric consistency loss** that guides global optimization by aligning predicted motion-induced brightness changes with event-observed measurements, improving spatiotemporal coherence under low light.

- We validate EAG3R across multiple challenging 3D vision tasks—including monocular depth estimation, camera pose tracking, and dynamic reconstruction—and show it significantly outperforms existing RGB-based pose-free methods, even under zero-shot nighttime conditions.

## 2 Related Work

**SfM and SLAM** Traditional Structure-from-Motion (SfM) [1, 44, 45, 50, 53, 54] and Simultaneous Localization and Mapping (SLAM) [11, 15, 38, 41] methods estimate 3D structure and camera motion by establishing 2D correspondences [5, 12, 36, 38, 49] or minimizing photometric errors [14, 15], followed by bundle adjustment (BA) [2, 6, 61]. While effective with dense views, these often struggle with sparse or ill-conditioned data. Recent learning-based approaches aim to improve robustness and efficiency: DUSt3R [64], directly regress dense point maps from image pairs using Transformer architectures [13] trained on large-scale 3D datasets. However, DUSt3R and its variants

[35, 39, 56, 59, 60] are primarily designed for static scenes and their performance degrades with dynamic content.

**Pose-free Dynamic Scene Reconstruction**    Reconstructing dynamic scenes without known poses is a core challenge in SLAM. Classical methods rely on joint pose estimation and dynamic region filtering via semantics [71] or optical flow [75], but depend heavily on accurate segmentation and tracking. Some methods estimate temporally consistent depth using geometric constraints [37] or generative priors [24, 51], yet suffer from fragmented reconstructions due to missing camera trajectories. Recent works jointly optimize depth and pose by refining pretrained depth models [46] using flow [65] and masks [23], e.g., in CasualSAM [74], Robust-CVD [30], and MegaSAM [32], the latter integrating DROID-SLAM [58] and diverse priors [43, 69]. Recent approaches employ direct pointmap regression, including MonST3R [72], DAS3R [67], CUT3R [62], and Easi3R [10], which leverage optical flow, segmentation, or attention for motion disentanglement.

**Low-light Enhancement**    Low-light image enhancement (LIE) aims to improve image quality under poor illumination. Traditional methods like histogram equalization [3] and Retinex-based algorithms [21] have limited adaptability, while deep learning approaches [66, 7, 63] achieve better results but still struggle in extreme darkness. Event cameras, with high dynamic range and temporal resolution, enable structural information preservation in very low light [76, 73], inspiring event-guided LIE methods [26, 34]. However, robust fusion of frame and event data under noise remains challenging. EvLight [33] addresses this with adaptive event-image feature fusion.

**Event-based 3D Vision**    Event cameras have enabled progress in 3D reconstruction under challenging lighting and fast motion [17]. Early work used stereo setups for disparity and multi-view stereo [8, 42, 77], followed by monocular methods based on geometric priors like camera trajectories [28, 47]. Recent approaches apply deep learning to stereo [40] and monocular [4, 9] settings, producing dense outputs such as meshes or voxels. Multimodal fusion with structured light [31] or RGB-D sensors [79] further improves robustness. Latest advances adapt NeRF [29, 48] and Gaussian Splatting [22, 25] to event data, enabling high-fidelity scene reconstruction and novel view synthesis.

## 3    Methods

**Overview**    Figure 1 shows the overall pipeline of EAG3R. Our work addresses the critical challenge of robust monocular 3D scene reconstruction—encompassing dynamic geometry, camera pose, and depth estimation—under extreme lighting conditions where traditional RGB-based methods often fail. EAG3R enhances the MonST3R framework by synergistically integrating standard RGB video frames $\{I^t \in \mathbb{R}^{H \times W \times 3}\}$ with asynchronous event streams $\{\mathcal{E}^t\}$ (sequences of $e_k = (x_k, y_k, t_k, p_k)$). This is achieved through two primary strategies: adaptive event-image feature fusion guided by signal quality, and an event-augmented global optimization that incorporates event-derived cues for static region masking and consistency loss.

The subsequent sections provide a detailed exposition: Section 3.1 reviews the foundational DUSt3R and MonST3R architectures. Section 3.2 then describes our *Event-data Integration and Feature Fusion* approach, including techniques for RGB image enhancement, the design of a lightweight event adapter, and our core SNR-aware fusion mechanism. Finally, Section 3.3 details the *Global Optimization with Event Consistency*, explaining how event-based consistency loss are integrated to achieve enhanced spatio-temporal coherence across both RGB and event data.

### 3.1    Preliminary

Our work builds upon DUSt3R and its dynamic extension MonST3R, which employ *pointmaps* for direct, dense 3D geometry estimation from images, facilitating pose-free monocular reconstruction.

**Pointmap-based Static Reconstruction.**    DUSt3R utilizes pointmaps ($X_{\mathrm{pm}} \in \mathbb{R}^{H \times W \times 3}$), assigning a 3D coordinate per pixel, predicted for image pairs ($I^a, I^b$) by a Transformer model:

$$(X_{\mathrm{pm}}^{a \to a}, X_{\mathrm{pm}}^{b \to a}) = \mathrm{Model}(I^a, I^b). \tag{1}$$

These encode relative geometry for depth and pose estimation and are refined via global optimization for multi-view consistency into a global point cloud $X^*$.

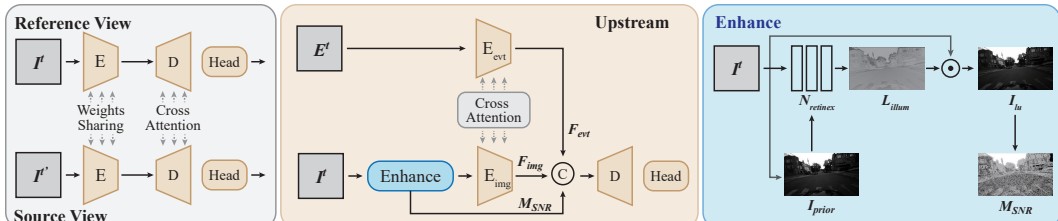

Figure 2: **EAG3R network.** Left: The DUSt3R (MonST3R) architecture with reference and source views processed via ViT encoder-decoder structure. Middle: Our method (only the upstream branch for the reference image is shown), which includes a lightweight event encoder and fuses event and image features with cross-attention. Right: The Retinex-based enhancement module estimates an illumination map and an SNR confidence map to guide adaptive fusion.

**Extension to Dynamic Scenes.** MonST3R adapts this for dynamic scenes by finetuning DUSt3R on dynamic datasets, predicting per-frame pointmaps. It optimizes a global scene model $X^*_{\text{global}}$ (comprising per-frame camera poses $\{P^t\}$, intrinsics $\{K^t\}$, and depth maps $\{D^t\}$) using an objective function:

$$\mathcal{L}_{\text{MonST3R}}(X^*_{\text{global}}) = \mathcal{L}_{\text{align}} + w_{\text{smooth}}\mathcal{L}_{\text{smooth}} + w_{\text{flow}}\mathcal{L}_{\text{flow}}, \tag{2}$$

guided by alignment, trajectory smoothness, and image-based optical flow ($\mathcal{L}_{\text{flow}}$) terms.

However, the reliance of these image-based methods on clear visual information causes them to struggle in low-light settings: RGB images $I^t$ lose crucial detail, and MonST3R's flow estimation with RAFT [57] can become unstable. Our EAG3R addresses these limitations by integrating asynchronous event data $E^t$ into both the feature extraction and global optimization stages, aiming for robust 3D reconstruction performance even in such challenging lighting conditions.

## 3.2 Event-data Integration and Feature Fusion

To enable robust geometry estimation in low-light scenarios, we redesign the encoding pipeline with a hybrid event-image architecture as is illustrated in Fig. 2. Our improvements begin with a Retinex-inspired enhancement module that operates on the raw input image $I^t$ to recover visibility in underexposed regions. This module also estimates a SNR map, $\mathcal{M}^t_{\text{snr}}$, which serves as a spatial prior for confidence-aware fusion. Next, we introduce a lightweight event adapter based on a Swin Transformer backbone, designed to extract high-fidelity features from the sparse event stream $\mathcal{E}^t$. We also establish cross-modal interaction through a cross-attention mechanism between event and image features. Finally, we propose an SNR-aware fusion strategy that adaptively balances image and event features based on local SNR, favoring images in well-lit areas and events in low-visibility regions. This yields a more informative and robust representation $\mathcal{F}^t$ for downstream 3D reconstruction.

**Retinex-based Image Enhancement.** To enhance image visibility under low-light conditions and provide a spatial reliability prior, we introduce a lightweight Retinex-inspired [7] enhancement module. Given an input image $I^t$, we estimate an illumination map $L^t_{\text{illum}}$ using a shallow network $\mathcal{N}_{\text{retinex}}$ with inputs $I^t$ and its channel-wise maximum projection $I^t_{\text{prior}} = \max_c(I^t)$, and compute the enhanced image via element-wise multiplication:

$$L^t_{\text{illum}} = \mathcal{N}_{\text{retinex}}(I^t, I^t_{\text{prior}}), \quad I^t_{\text{lu}} = I^t \odot L^t_{\text{illum}}. \tag{3}$$

To guide adaptive fusion, we compute a SNR map $\mathcal{M}^t_{\text{snr}}$ indicating local image reliability. We convert $I^t_{\text{lu}}$ to grayscale $I^t_g$, apply mean filtering to obtain $\widetilde{I}^t_g$, and define:

$$\mathcal{M}^t_{\text{snr}} = \frac{\widetilde{I}^t_g}{\left|I^t_g - \widetilde{I}^t_g\right| + \epsilon}, \tag{4}$$

where $\epsilon$ ensures numerical stability. This SNR map emphasizes high-confidence regions and suppresses noise-dominated areas, enabling reliability-aware feature fusion downstream.

**Lightweight Event Adapter.** To effectively harness asynchronous event streams $\mathcal{E}^t$ for dense geometric prediction, we introduce a lightweight event adapter by employing a Swin Transformer backbone initialized with weights from a self-supervised, context-based pre-training regimen on event data [70]. The input events from $\mathcal{E}^t$ are voxelized into a spatiotemporal grid and processed by the pre-trained Swin Transformer encoder, yielding hierarchical event features $\{F_{\text{evt},l}^t\}_{l=1}^4$. Corresponding hierarchical image features $\{F_{\text{img},l}^t\}_{l=1}^4$ are extracted at every 6 layers from intermediate representations within the pre-trained image encoder. At each hierarchical stage $l$, the event features $F_{\text{evt},l}^t$ are spatially aligned and dimension-matched with their respective image counterparts $F_{\text{img},l}^t$.

We then apply cross-attention[68], using event features as queries and image features as keys and values:

$$F_{\text{evt},l}' = \text{CrossAttn}(Q = F_{\text{evt},l}^t, \ K = F_{\text{img},l}^t, \ V = F_{\text{img},l}^t), \tag{5}$$

Importantly, the image encoder remains frozen, and only the event adapter is updated. This training strategy ensures efficient adaptation without disrupting the pretrained image backbone, while allowing the event pathway to learn to compensate for degraded or missing visual cues.

**SNR-aware Feature Aggregation.** We combine the final features from the image ($F_{\text{img-final}}^t$) and event ($F_{\text{evt-final}}'^t$) encoders into a unified representation $\mathcal{F}^t$, guided by the normalized SNR map $\hat{\mathcal{M}}_{\text{snr}}^t$. Specifically, we weight image features by $\hat{\mathcal{M}}_{\text{snr}}^t$ and event features by its complement $(1 - \hat{\mathcal{M}}_{\text{snr}}^t)$, followed by concatenation:

$$F_{\text{cat}}^t = (F_{\text{img-final}}^t \odot \hat{\mathcal{M}}_{\text{snr}}^t) \ \| \ (F_{\text{evt-final}}'^t \odot (1 - \hat{\mathcal{M}}_{\text{snr}}^t)), \tag{6}$$

where $\odot$ denotes element-wise multiplication with channel-wise broadcasting. The concatenated features $F_{\text{cat}}^t$ undergo a projection to match the input dimensionality of the downstream decoder.

This adaptive feature aggregation dynamically prioritizes image features under high-SNR conditions and event features under low-light scenarios, yielding robust and illumination-invariant representations for effective downstream 3D reconstruction.

## 3.3 Event-Enhanced Global Optimization

To enhance the performance of MonST3R's 3D reconstruction and camera pose estimation, particularly in challenging low-light environments where image-based cues are compromised, our approach augments its global optimization framework. The primary enhancement is the introduction of an event-based photometric consistency loss, $\mathcal{L}_{\text{event}}$. This integration leverages the inherent advantages of event cameras– such as their high dynamic range and ability to capture dynamics even with minimal illumination– to provide robust supervisory signals. The subsequent sections detail the formulation of this $\mathcal{L}_{\text{event}}$ from raw event data (Section 3.3.1) and its incorporation into the joint optimization process (Section 3.3.2).

### 3.3.1 Event-Based Photometric Consistency Loss

The event-based photometric consistency loss, $\mathcal{L}_{\text{event}}$, is formulated to evaluate the alignment of brightness change patterns observed within salient image patches. It achieves this by comparing two distinct representations of these patterns: the first, $\Delta L_{\mathcal{P}_m}(u)$, is derived directly from the raw event stream $\mathcal{E}$; the second, $\Delta \hat{L}_{\mathcal{P}_m}(u; X_{\text{global}})$, is synthesized by integrating photometric information from an intensity image with scene motion inferred from the global state estimate $X_{\text{global}}$. The process of computing this loss is visualized in Fig. 3.

**Observed Brightness Increments from Events:** Given an event stream $\mathcal{E}$ corresponding to the time interval between frames $I^t$ and $I^{t'}$, we compute the observed brightness increment within each salient image patch $\mathcal{P}_m$ by aggregating events $e_k = (x_k, y_k, t_k, p_k)$ occurring in both space and time over the patch and interval $\Delta\tau = t' - t$:

$$\Delta L_{\mathcal{P}_m}(u) = \sum_{t_j \in [t,t'], \ (x_j,y_j) \in \mathcal{P}_m} p_j \delta(u - u_j), \tag{7}$$

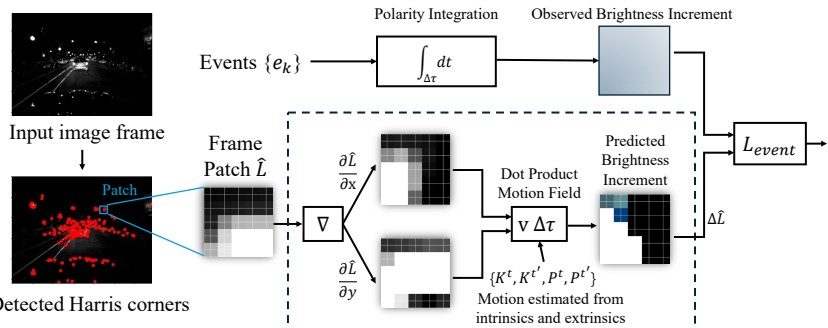

Figure 3: **Event-based photometric consistency loss.** Harris corners are detected on the input image to define salient patches. Observed brightness increments are computed by integrating event polarities, while predicted increments are synthesized from image gradients and motion. The loss $\mathcal{L}_{\text{event}}$ measures their alignment.

where $u = (x, y)$ denotes local coordinates within the patch. Each patch $\mathcal{P}_m$ is centered at a Harris corner detected on the reference intensity image $\mathcal{I}^t$ and covers a small spatial neighborhood around the corner location. This ensures that the selected regions exhibit strong intensity gradients and are thus well-suited for event-based tracking. The aggregation in Equation (7) yields a polarity-weighted event accumulation image representing the measured brightness changes within each patch.

**Brightness Increment Model from Intensity and Motion:** To estimate the brightness change within a salient image patch $\mathcal{P}_m$, we adopt a predictive model derived from the principle of brightness constancy. This model synthesizes the expected brightness increment $\Delta \hat{L}_{\mathcal{P}_m}(u; X_{\text{global}})$ by combining photometric and geometric cues, specifically:

- **Local Intensity Gradient:** The spatial gradient $\frac{\partial \mathcal{I}^t_{\text{grad}}}{\partial u}(\hat{u})|_{\mathcal{P}_m}$ is computed over the patch $\mathcal{P}_m$ from the intensity image $\mathcal{I}^t$ at time $t$.
- **Inter-frame Pixel Motion:** The motion field $\Delta u^{t \to t'}_{cam}(\hat{u}, X_{\text{global}})$ represents the per-pixel displacement between frames $t$ and $t'$, computed by projecting 3D points using the depth map $D^t$ and the camera intrinsics and extrinsics $(K^t, K^{t'}, P^t, P^{t'})$ contained in the global state $X_{\text{global}}$.

Assuming locally constant optical flow and small inter-frame displacements, the predicted brightness increment is expressed as:

$$\Delta \hat{L}_{\mathcal{P}_m}(\hat{u}; X_{\text{global}}) = -\frac{\partial \mathcal{I}^t_{\text{grad}}}{\partial u}(\hat{u})|_{\mathcal{P}_m} \cdot \Delta u^{t \to t'}_{cam}(\hat{u}, X_{\text{global}})|_{\mathcal{P}_m} \cdot \Delta \tau \cdot C, \tag{8}$$

where $\Delta \tau$ denotes the integration interval and $C$ is the contrast sensitivity threshold intrinsic to the event sensor. This expression follows the generative model introduced in [20].

**Event-Based Loss Objective:** While Equation (8) provides an explicit formulation, the scale factor $\Delta \tau \cdot C$ is unknown and varies across sensors and operating conditions. To eliminate this ambiguity, we normalize both the observed and predicted brightness increment patches to unit $L^2$ norm before computing the residual. This yields an objective that is invariant to the unknown contrast scale and focuses solely on the alignment of gradient directions:

$$\mathcal{L}_{\text{event}}(X_{\text{global}}) = \sum_{\mathcal{P}_m} \sum_{u \in \mathcal{P}_m} \left\| \frac{\Delta L_{\mathcal{P}_m}(u)}{\sum_{u \in \mathcal{P}_m} \|\Delta L_{\mathcal{P}_m}(u)\|} - \frac{\Delta \hat{L}_{\mathcal{P}_m}(u; X_{\text{global}})}{\sum_{u \in \mathcal{P}_m} \|\Delta \hat{L}_{\mathcal{P}_m}(u; X_{\text{global}})\|} \right\|^2. \tag{9}$$

This loss enforces that brightness variations predicted from image gradients and estimated motion are consistent with real event stream observations, thereby providing a principled supervision signal for optimizing $X_{\text{global}}$.

### 3.3.2 Joint Optimization with Event-Based Constraints

The event-based loss $\mathcal{L}_{\text{event}}$ is integrated into MonST3R's global optimization objective (Eq. 2). The augmented objective to find the optimal global scene model $X_{\text{global}}^*$, pairwise alignments $\{P_W^*\}$, and scales $\{\sigma^*\}$ becomes:

$$X_{\text{global}}^*, \{P_W^*\}, \{\sigma^*\} = \arg \min_{X_{\text{global}}, \{P_W\}, \{\sigma\}} \Big( \mathcal{L}_{\text{align}}(X_{\text{global}}, \{P_W\}, \{\sigma\}) + w_{\text{smooth}} \mathcal{L}_{\text{smooth}}(X_{\text{global}})$$
$$+ w_{\text{flow}} \mathcal{L}_{\text{flow}}(X_{\text{global}}) + w_{\text{event}} \mathcal{L}_{\text{event}}(X_{\text{global}}) \Big) \tag{10}$$

where $w_{\text{event}}$ is the $w_{\text{event\_base}}$ scaled by the mean of $(1 - S_{\text{norm}})$, where $S_{\text{norm}}$ are the normalized corner SNR values.

$\mathcal{L}_{\text{event}}$ provides a more dependable constraint on geometry and motion by leveraging informative event patterns in salient patches. Minimizing this augmented objective refines the state estimate $X_{\text{global}}$ by ensuring that brightness changes modeled from intensity and motion, $\Delta \hat{L}_{\mathcal{P}_m}(u; X_{\text{global}})$, align closely with observed event patterns $\Delta L_{\mathcal{P}_m}(u)$. This synergy enhances the accuracy and robustness of 3D reconstruction and pose estimation, particularly when conventional image quality is poor.

## 4 Experiments

We evaluate our method in a variety of tasks, including depth estimation (Section 4.2), camera pose estimation (Section 4.3) and 4D reconstruction (Section 4.4). We perform ablation studies in Section 4.5. We compare EAG3R with state-of-the-art pose free learning-based reconstruction method, including DUSt3R [64], MonST3R [72], and Easi3R [10].

### 4.1 Experiment Details

For training, we fine-tune the MonST3R baseline by training its ViT-Base decoder, DPT heads, Enhancement Net, and the Event Adapter. The Event Adapter is pre-trained on the ETartanAir dataset. Fine-tuning is performed for 25 epochs, using 8,000 image-event pairs per epoch. We employ the AdamW optimizer with a learning rate of $5 \times 10^{-5}$ and a mini-batch size of 4 per GPU. The training process completes in approximately 24 hours on 4 NVIDIA RTX 3090 GPUs. For global optimization, we adopt the same setting as MonST3R, with hyperparameters $w_{smooth} = 0.01$, $w_{flow} = 0.01$, and $w_{event\_base} = 0.01$. We use the Adam optimizer for 300 iterations with a learning rate of 0.01.

For dataset selection, we initially attempted to fine-tune the MonST3R baseline using events generated via Video-to-Events (V2E)[19] from MonST3R's fine-tuning datasets. However, the noise in V2E-generated events led to gradient explosion during training, prompting us to switch to datasets with real event captures and ground truth (GT) depth. Given the scarcity of such data, we selected the Multi Vehicle Stereo Event Camera (MVSEC) dataset [78]. It provides synchronized stereo events and reliable LiDAR-derived depth GT. To ensure a fair zero-shot evaluation of low-light performance, all models were trained exclusively on MVSEC's `outdoor_day2` sequence (normal daylight) and tested on the challenging `outdoor_night1-3` sequences (extreme low-light).

### 4.2 Monocular Depth Estimation

We evaluate monocular depth estimation on the MVSEC `outdoor_night1-3` sequences, which feature extreme low-light conditions with significant noise and underexposure. All models are trained solely on the MVSEC `outdoor_day2` sequence and tested zero-shot on these nighttime scenes to ensure a fair comparison. We report results using standard metrics: Absolute Relative Error (Abs Rel ↓), Scale-invariant RMSE log (RMSE log ↓), and the threshold accuracy $\delta < 1.25$ (↑), where lower is better for error metrics and higher is better for accuracy.

As shown in Tab. 1, DUSt3R performs poorly due to the severe degradation of visual cues at night. However, applying RetinexFormer, a widely used image enhancement network, as a preprocessing light-up step (denoted as (LightUp)) does not yield significant improvements and, in some cases, degrades performance, indicating that image enhancement alone is insufficient for this task without joint optimization with the downstream model. Fine-tuning MonST3R improves its performance across

Table 1: **Monocular depth estimation performance on nighttime scenes.** Evaluation is conducted on the MVSEC `Night1`, `Night2`, and `Night3` sequences. Standard metrics are used: Abs Rel $\downarrow$, $\delta < 1.25 \uparrow$, and RMSE log $\downarrow$. Best performing method in **bold**, second best underlined.

| Method | Night1 | | | Night2 | | | Night3 | | |
|---|---|---|---|---|---|---|---|---|---|
| | Abs Rel $\downarrow$ | $\delta<1.25 \uparrow$ | RMSE log $\downarrow$ | Abs Rel $\downarrow$ | $\delta<1.25 \uparrow$ | RMSE log $\downarrow$ | Abs Rel $\downarrow$ | $\delta<1.25 \uparrow$ | RMSE log $\downarrow$ |
| DUSt3R[64] | 0.407 | 0.393 | 0.534 | 0.415 | 0.384 | 0.495 | 0.463 | 0.335 | 0.534 |
| MonST3R[72] | 0.370 | 0.373 | 0.497 | 0.309 | 0.469 | 0.409 | 0.317 | 0.453 | 0.418 |
| DUSt3R (LightUp) | 0.425 | 0.351 | 0.548 | 0.462 | 0.347 | 0.537 | 0.525 | 0.293 | 0.592 |
| MonST3R (LightUp) | 0.370 | 0.369 | 0.503 | 0.316 | 0.451 | 0.431 | 0.329 | 0.441 | 0.444 |
| MonST3R (Finetune) | 0.376 | 0.426 | 0.478 | 0.328 | 0.472 | 0.415 | 0.302 | 0.509 | 0.401 |
| **EAG3R** | **0.353** | **0.491** | **0.416** | **0.307** | **0.518** | **0.383** | **0.288** | **0.533** | **0.371** |

Table 2: **Camera pose estimation on all MVSEC nighttime sequences.** Evaluation is conducted on the MVSEC `Night1`, `Night2`, and `Night3` sequences. Standard metrics are used: ATE $\downarrow$, RPE trans $\downarrow$, and RPE rot $\downarrow$. Best performing method in **bold**, second best underlined.

| Method | Night1 | | | Night2 | | | Night3 | | |
|---|---|---|---|---|---|---|---|---|---|
| | ATE $\downarrow$ | RPE trans $\downarrow$ | RPE rot $\downarrow$ | ATE $\downarrow$ | RPE trans $\downarrow$ | RPE rot $\downarrow$ | ATE $\downarrow$ | RPE trans $\downarrow$ | RPE rot $\downarrow$ |
| DUSt3R[64] | 1.474 | 0.914 | 2.995 | 3.921 | 2.207 | 10.761 | 4.109 | 2.401 | 11.309 |
| MonST3R[72] | 0.559 | 0.317 | 0.369 | 0.626 | 0.341 | 1.460 | 0.733 | 0.427 | 1.371 |
| Easi3R$_{dust3r}$[10] | 1.505 | 0.953 | 3.024 | 3.884 | 2.205 | 10.608 | 4.102 | 2.398 | 11.306 |
| Easi3R$_{monst3r}$[10] | 0.550 | 0.303 | 0.362 | 0.606 | 0.328 | 1.462 | 0.712 | 0.414 | 1.369 |
| MonST3R (Finetune) | 0.580 | 0.284 | 0.214 | 0.467 | 0.210 | 0.374 | 0.402 | 0.183 | 0.370 |
| Easi3R$_{monst3r}$ (Finetune) | 0.540 | 0.263 | 0.214 | 0.448 | **0.202** | 0.374 | **0.394** | **0.178** | 0.371 |
| **EAG3R (Ours)** | **0.482** | **0.201** | **0.143** | **0.428** | 0.207 | **0.342** | 0.409 | 0.201 | **0.320** |

most metrics, demonstrating the benefit of domain adaptation. Our method, EAG3R, outperforms all baselines across all three nighttime sequences, indicating both accurate and reliable depth predictions. EAG3R leverages asynchronous event signals that remain informative in such challenging low-light settings, which enables strong generalization capabilities despite the model never having been trained on nighttime data. These results highlight the distinct advantage of incorporating event-based cues for robust depth estimation under extreme illumination conditions, where conventional RGB-based methods—even when augmented with fine-tuning or pre-enhancement—struggle to perform reliably.

## 4.3 Camera Pose Estimation

We evaluate camera pose estimation on the challenging MVSEC nighttime sequences using standard metrics (ATE, RPE trans, RPE rot; lower is better), following a consistent zero-shot protocol (trained on `outdoor_day2`) as in our depth experiments.

As shown in Tab. 2, RGB-only baselines such as DUSt3R fail under extreme low-light conditions, while MonST3R offers improved results. Fine-tuning MonST3R leads to substantial gains, particularly in RPE trans and RPE rot, with further improvements from Easi3R variants. Despite these enhancements, our proposed EAG3R consistently achieves the best performance across most metrics and sequences. This advantage comes from EAG3R's effective use of asynchronous event data, which provides reliable motion cues even when RGB inputs are heavily degraded. As a result, EAG3R maintains robust tracking and delivers more accurate camera trajectories, highlighting the strength of event-based sensing in scenarios where conventional methods often fail. As illustrated in Fig. 4, the predicted trajectory from EAG3R exhibits lower drift and aligns more closely with the ground truth compared to DUSt3R and MonST3R, further demonstrating its superiority in precise pose estimation.

## 4.4 Dynamic Reconstruction

We evaluate dynamic 3D reconstruction on the MVSEC `outdoor_night1-3` sequences. Prior methods such as DUSt3R and MonST3R serve as RGB-based baselines, with MonST3R extending pointmap prediction to dynamic scenes and Easi3R variants incorporating motion-aware masking. However, all remain limited under low-light conditions due to their reliance on degraded RGB inputs.

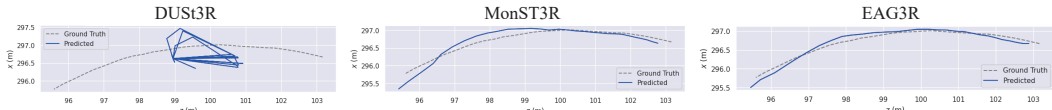

Figure 4: **Comparison of estimated camera trajectories.** The predicted trajectories (solid blue) from DUS3R, MonST3R, and EAG3R are evaluated against the ground truth (dashed gray). Notably, EAG3R demonstrates a trajectory that more closely aligns with the ground truth.

Table 3: **Ablation study on depth estimation performance on the `Night3` sequence.** Modules are incrementally added to the MonST3R baseline. Each addition improves performance, with the full EAG3R system achieving the best results.

| Method | Abs Rel ↓ | $\delta < 1.25$ ↑ | RMSE log ↓ |
|---|---|---|---|
| MonST3R (Baseline) | 0.317 | 0.453 | 0.418 |
| MonST3R (Finetune) | 0.302 | 0.509 | 0.401 |
| + Event | 0.297 | 0.518 | 0.396 |
| + Event + LightUp | 0.291 | 0.523 | 0.388 |
| **+ Event + LightUp + SNR Fusion (Full)** | **0.288** | **0.533** | **0.371** |

In contrast, EAG3R directly integrates asynchronous event streams into the 4D reconstruction pipeline, allowing for improved motion handling and robustness to illumination changes. Qualitative results, provided in the appendix, show that EAG3R produces cleaner, more complete reconstructions and better preserves dynamic scene details compared to purely frame-based methods.

### 4.5 Ablation Study

To better understand the contribution of each design component in EAG3R, we conduct a systematic ablation study on the MVSEC `outdoor_night3` sequence for monocular depth estimation. Starting from the MonST3R baseline, we incrementally add our proposed modules: event inputs, the LightUp enhancement network, and the SNR-aware fusion mechanism. Results are shown in Tab. 3.

Each component contributes positively to the final performance. The introduction of event streams already leads to a substantial improvement, validating the value of asynchronous visual signals in low-light scenarios. Incorporating the LightUp module provides additional gains by improving the quality of underexposed RGB inputs. Finally, the SNR-guided fusion further boosts robustness by adaptively emphasizing reliable features from either modality, particularly in noisy or degraded regions. The combination of these modules leads to the strongest performance, confirming the effectiveness of our full EAG3R design.

## 5 Conclusion

We presented **EAG3R**, a event-augmented framework for robust 3D geometry estimation under dynamic and low-light conditions. Built on the MonST3R backbone, EAG3R introduces a lightweight event adapter and a retinex-inspired light-up module, an SNR-aware fusion mechanism, and an event-based photometric consistency loss. These components enable reliable depth and pose estimation where conventional RGB-only methods struggle. EAG3R achieves strong zero-shot generalization to nighttime scenes, consistently outperforming state-of-the-art baselines in depth, camera pose estimation, and dynamic reconstruction tasks. Our results highlight the value of integrating asynchronous event signals into geometry pipelines. We discuss limitations and broader impact in the appendix.

## 6 Acknowledgements

This work has been supported by the National Key R&D Program of China (Grant No. 2022YFB3608300), Hong Kong Research Grant Council - Early Career Scheme (Grant No. 27209621), General Research Fund Scheme (Grant No. 17202422, 17212923, 17215025), Theme-based Research (Grant No. T45-701/22-R) and Shenzhen Science and Technology Innovation

Commission (SGDX20220530111405040). Part of the described research work is conducted in the JC STEM Lab of Robotics for Soft Materials funded by The Hong Kong Jockey Club Charities Trust. This research was partially conducted by ACCESS – AI Chip Center for Emerging Smart Systems, supported by the InnoHK initiative of the Innovation and Technology Commission of the Hong Kong Special Administrative Region Government.

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

# A  Technical Appendices and Supplementary Material

## A.1  Dataset Processing

The Multi-Vehicle Stereo Event Camera (MVSEC) dataset integrates three distinct sensor modalities, each with independent timestamps: Active Pixel Sensor (APS) for frame-based images, Dynamic Vision Sensor (DVS) for event-based data, and Velodyne Puck LITE for depth measurements. These modalities operate at different frequencies, necessitating careful synchronization to ensure data consistency. The Velodyne Puck LITE provides depth data at a fixed frequency of 20 Hz, while the APS captures frames at approximately 100 Hz during daytime and 10 Hz during nighttime. The DVS generates asynchronous event streams, which are temporally aggregated into voxel grids for processing.

To align these modalities, we project depth measurements to image timestamps and voxelize event data between consecutive timestamps. The alignment process varies between daytime and nighttime sequences due to differences in frame frequency relative to depth data.

### A.1.1  Daytime Sequence Processing

For daytime sequences, where APS operates at 100 Hz, we associate each depth ground truth from the Velodyne Puck LITE with the temporally closest APS frame. This is achieved through the following steps:

1. **Pose Interpolation for Images**: Interpolate camera poses at all APS image timestamps $t_i$. Given discrete pose measurements $P(t_k)$ at times $t_k$, we compute the interpolated pose $P(t_i)$ at image timestamp $t_i$ using linear interpolation for translation and spherical linear interpolation (SLERP) for rotation:

$$P(t_i) = P(t_k) + \frac{t_i - t_k}{t_{k+1} - t_k}(P(t_{k+1}) - P(t_k)),$$

   where $P(t) = (R(t), T(t))$ represents the rotation $R(t) \in SO(3)$ and translation $T(t) \in \mathbb{R}^3$.

2. **Pose Interpolation for Depth**: Similarly, interpolate poses at depth timestamps $t_d$ to obtain $P(t_d)$, using the same interpolation method as above.

3. **Timestamp Matching**: For each depth timestamp $t_d$, identify the nearest image timestamp $t_i$ by minimizing the temporal difference:

$$t_i = \arg \min_{t_j \in T_I} |t_d - t_j|,$$

   where $T_I$ is the set of all image timestamps.

4. **Depth Warping**: Warp the depth data to the selected image timestamp $t_i$ using the relative transformation between poses $P(t_d)$ and $P(t_i)$. For a 3D point $X_d$ in the depth sensor's coordinate frame at $t_d$, the warped point $X_i$ at $t_i$ is computed as:

$$X_i = R(t_i)R(t_d)^{-1}(X_d - T(t_d)) + T(t_i).$$

### A.1.2  Nighttime Sequence Processing

During nighttime sequences, the APS frame rate drops to 10 Hz, resulting in multiple depth measurements (at 20 Hz) per APS frame. We process the depth data as follows:

1. **Depth Projection to 3D**: Project all depth measurements within the temporal window of a single APS frame into a 3D point cloud. For a depth measurement $d$ at timestamp $t_d$, the 3D point $X_d$ is obtained using the sensor's intrinsic calibration and pose $P(t_d)$:

$$X_d = P(t_d) \cdot \text{unproject}(d),$$

   where unproject converts the depth measurement to a 3D point in the sensor's local frame.

2. **Selection of Closest Depth**: For each APS frame at timestamp $t_i$, select the depth measurement from the 3D point cloud that is temporally closest to $t_i$, as determined by the minimum temporal difference $|t_d - t_i|$.

Table A.1: **Video depth estimation performance on nighttime scenes.** Evaluation is conducted on the MVSEC `Night1`, `Night2`, and `Night3` sequences. Standard metrics are used: Abs Rel ↓, $\delta < 1.25$ ↑, and Log RMSE ↓. Best performing method in **bold**, second best underlined.

| Method | Night1 | | | Night2 | | | Night3 | | |
|---|---|---|---|---|---|---|---|---|---|
| | Abs Rel ↓ | $\delta$<1.25 ↑ | Log RMSE ↓ | Abs Rel ↓ | $\delta$<1.25 ↑ | Log RMSE ↓ | Abs Rel ↓ | $\delta$<1.25 ↑ | Log RMSE ↓ |
| DUSt3R[64] | 0.432 | 0.374 | 0.547 | 0.410 | 0.397 | 0.493 | 0.510 | 0.322 | 0.554 |
| MonST3R[72] | 0.380 | 0.388 | 0.486 | **0.299** | 0.494 | **0.388** | **0.296** | 0.499 | 0.392 |
| Easi3R$_{dust3r}$[10] | 0.427 | 0.388 | 0.549 | 0.435 | 0.376 | 0.515 | 0.504 | 0.324 | 0.566 |
| Easi3R$_{monst3r}$[10] | 0.375 | 0.381 | 0.484 | 0.308 | 0.490 | 0.397 | 0.314 | 0.465 | 0.404 |
| **EAG3R (Ours)** | **0.357** | **0.482** | **0.427** | 0.321 | **0.494** | 0.402 | 0.302 | **0.512** | **0.302** |

### A.1.3 Rectification and Hole Filling

To enhance data quality, we use rectified APS frames, DVS event data, and depth measurements. Rectification of APS frames introduces sparse regions ("holes") due to the transformation process. We address this by applying spatial interpolation to fill these holes, ensuring a continuous image. Specifically, for a pixel $(x, y)$ in a sparse region, we estimate its value $I(x, y)$ using a weighted average of neighboring valid pixels:

$$I(x, y) = \frac{\sum_{(x', y') \in N} w(x', y') I(x', y')}{\sum_{(x', y') \in N} w(x', y')},$$

where $N$ is the set of neighboring valid pixels, and $w(x', y')$ is a distance-based weight (e.g., inverse Euclidean distance). This ensures the rectified frames are suitable for downstream tasks such as scene understanding and 3D reconstruction.

This processing pipeline ensures robust temporal and spatial alignment across the APS, DVS, and depth modalities, enabling effective utilization of the MVSEC dataset.

## A.2 Video Depth Estimation Results on MVSEC

We evaluate video depth estimation under extreme low-light conditions using the MVSEC `outdoor_night1`, `outdoor_night2`, and `outdoor_night3` sequences. All models are trained solely on the `outdoor_day2` sequence and tested in a zero-shot nighttime setting. Following standard protocol, we report Absolute Relative Error (Abs Rel ↓), RMSE log ↓, and $\delta < 1.25$ (↑) over all frames in each sequence.

As shown in Tab. A.1, conventional RGB-based baselines such as DUSt3R and MonST3R suffer from errors due to degraded visual signals at night, and even Easi3R exhibit limited temporal consistency. In contrast, EAG3R achieves superior performance across all sequences and metrics. By combining RGB-based pointmaps with temporally precise event cues, EAG3R effectively preserves geometric detail and improves depth stability over time. The event-based supervision introduces an additional constraint on spatiotemporal coherence, which regularizes the optimization even when image content is weak or noisy. These results demonstrate that augmenting pointmap-based reconstruction with event streams enables robust, temporally-consistent video depth estimation in dynamic and low-light environments.

## A.3 Dynamic Reconstruction Results

We present dynamic 4D reconstruction results of EAG3R on challenging real-world sequences involving fast-moving objects and adverse lighting. As illustrated in Fig. A.1, our method produces temporally coherent and geometrically accurate 3D point clouds, even under degraded RGB conditions and rapid scene changes.

Notably, in the highlighted sequence, a car passes through the middle of the scene—a moment where RGB-based methods fail to produce stable or even visible reconstructions due to motion blur and low illumination. EAG3R, powered by event-based cues, successfully reconstructs the moving vehicle with sharp geometry and consistent motion across frames. This showcases the unique advantage of leveraging asynchronous event data for robust dynamic scene understanding.

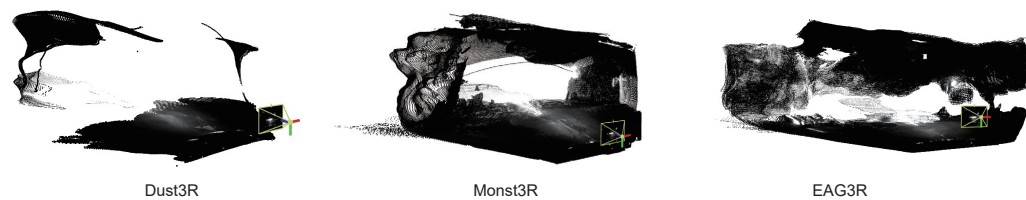

| Dust3R | Monst3R | EAG3R |

Figure A.1: Qualitative comparison on dynamic scenes. Our method reconstructs consistent 3D geometry even when a fast-moving vehicle passes through the scene. RGB-only methods fail to capture this motion reliably.

## A.4    Summary of Existing Event-RGB Datasets

Our choice of the MVSEC dataset was guided by the strict requirements of our task: robust 3D geometry estimation in dynamic scenes under extreme lighting. This demands datasets with aligned RGB, event data, and accurate ground truth for both depth and pose — a combination that is rarely available. As shown in Table A.2, few existing datasets satisfy the necessary conditions for evaluating dynamic reconstruction under challenging lighting.

Table A.2: Comparison of RGB-event datasets.

| Dataset | Low-light | Dynamic | RGB | Depth Sensor | GT Pose | Platform | Environment |
|---------|-----------|---------|-----|--------------|---------|----------|-------------|
| DSEC | ✓ | ✓ | ✓ | LiDAR-16 | ✗ | Car | Outdoor |
| UZH-FPV | ✗ | ✓ | ✓ | ✗ | MoCap | Drone | Indoor/Outdoor |
| DAVIS 240C | ✗ | ✓ | ✓ | ✗ | MoCap | Handheld | Indoor/Outdoor |
| GEN1 | ✓ | ✓ | ✓ | ✗ | ✗ | Car | Outdoor |
| Prophesee 1MP | ✓ | ✓ | ✓ | ✗ | ✗ | Car | Outdoor |
| TUM-VIE | ✓ | ✓ | ✓ | ✗ | MoCap | Handheld / Head-mounted | Indoor/Outdoor |
| MVSEC | ✓ | ✓ | ✓ | LiDAR-16 | MoCap / Cartographer | Car / Drone | Indoor/Outdoor |
| M3ED | ✓ | ✓ | ✓ | LiDAR-64 | LIO | Car / Legged Robot / Drone | Indoor/Outdoor |

## A.5    Ablations on Design Rationales

This section presents ablation studies analyzing the design rationales of **EAG3R**. Although EAG3R is built upon the pointmap-based reconstruction framework, it is the first to incorporate *asynchronous event streams*, enabling robust and pose-free 4D reconstruction under extreme lighting conditions. Three key design aspects are examined: the Retinex-inspired confidence estimation, the lightweight event adapter, and the multi-stage feature fusion.

### A.5.1    Addressing Extreme-Light Challenges for Geometric Estimation

A straightforward strategy for handling low-light conditions is to apply image enhancement as a preprocessing step. However, this approach is found to be suboptimal for geometric estimation due to introduced artifacts and the lack of structural consistency preservation.

As shown in Table A.3, applying RetinexFormer ("LightUp") before reconstruction provides limited or even negative benefit, indicating that conventional enhancement does not effectively support cross-modal fusion.

Table A.3: **Ablation on image enhancement under extreme lighting.** Evaluation is conducted on the MVSEC `Night1-3` sequences. Metrics: Abs Rel ↓, $\delta < 1.25$ ↑, RMSE log ↓. Best in **bold**.

| Method | Night1 | | | Night2 | | | Night3 | | |
|--------|--------|--|--|--------|--|--|--------|--|--|
| | Abs Rel ↓ | $\delta < 1.25$ ↑ | RMSE log ↓ | Abs Rel ↓ | $\delta < 1.25$ ↑ | RMSE log ↓ | Abs Rel ↓ | $\delta < 1.25$ ↑ | RMSE log ↓ |
| MonST3R | 0.370 | 0.373 | 0.497 | 0.309 | 0.469 | 0.409 | 0.317 | 0.453 | 0.418 |
| MonST3R (LightUp) | 0.370 | 0.369 | 0.503 | 0.316 | 0.451 | 0.431 | 0.329 | 0.441 | 0.444 |
| **EAG3R (Ours)** | **0.353** | **0.491** | **0.416** | **0.307** | **0.518** | **0.383** | **0.288** | **0.533** | **0.371** |

To address this limitation, EAG3R introduces a Retinex-inspired SNR estimation module that computes a spatially varying confidence map rather than directly enhancing images. This map guides

the adaptive fusion of RGB and event features, assigning higher confidence to events in dark or noisy regions and to RGB inputs where illumination is reliable. This constitutes, to the best of our knowledge, the first use of a Retinex-based confidence mechanism in event-guided geometric reconstruction.

### A.5.2 Efficient and Effective Event Adapter

Events are inherently sparse and asynchronous, posing challenges for efficient feature adaptation. A direct reuse of a heavy RGB encoder with zero-convolution initialization, leads to inefficient training and limited gains (as is compared in A.4).

Table A.4: **Ablation on event adapter design.** Comparison between zero-convolution initialization and the proposed lightweight Swin Transformer-based adapter. Evaluation is conducted on the MVSEC `Night1-3` sequences. Metrics: Abs Rel $\downarrow$, $\delta < 1.25 \uparrow$, RMSE log $\downarrow$.

| Method | Abs Rel $\downarrow$ | $\delta < 1.25 \uparrow$ | RMSE log $\downarrow$ |
|---|---|---|---|
| Zero_Conv | 0.377 / 0.323 / 0.302 | 0.446 / 0.478 / 0.485 | 0.449 / 0.399 / 0.379 |
| **EAG3R (Ours)** | **0.353 / 0.307 / 0.288** | **0.491 / 0.518 / 0.533** | **0.416 / 0.383 / 0.371** |

Our adapter is pretrained on large-scale event-only datasets using self-supervised objectives, enabling the transfer of general motion and edge priors. This design ensures fast convergence and strong generalization, particularly under data-scarce or extreme-light conditions.

### A.5.3 Robust and Principled Feature Fusion

RGB and event modalities differ substantially in spatial density, temporal resolution, and noise characteristics. Naive strategies, such as feature concatenation or single-layer attention, are insufficient to capture their complementary information.

Table A.5: **Ablation on fusion strategies.** Comparison between simple additive, single-layer attention, and the proposed multi-stage adaptive fusion. Evaluation is conducted on the MVSEC `Night1-3` sequences. Metrics: Abs Rel $\downarrow$, $\delta < 1.25 \uparrow$, RMSE log $\downarrow$.

| Method | Night1 | | | Night2 | | | Night3 | | |
|---|---|---|---|---|---|---|---|---|---|
| | Abs Rel $\downarrow$ | $\delta < 1.25 \uparrow$ | RMSE log $\downarrow$ | Abs Rel $\downarrow$ | $\delta < 1.25 \uparrow$ | RMSE log $\downarrow$ | Abs Rel $\downarrow$ | $\delta < 1.25 \uparrow$ | RMSE log $\downarrow$ |
| Feature Add | 0.357 | 0.482 | 0.417 | 0.312 | 0.519 | 0.384 | 0.295 | 0.535 | 0.373 |
| Last-layer Attention | 0.361 | 0.495 | 0.426 | 0.322 | 0.486 | 0.396 | 0.296 | 0.515 | 0.380 |
| **EAG3R (Ours)** | **0.353** | **0.491** | **0.416** | **0.307** | **0.518** | **0.383** | **0.288** | **0.533** | **0.371** |

EAG3R employs a multi-stage adaptive fusion mechanism that combines:

- **Cross-attention within the encoder**, where event features query multi-scale RGB features, enabling nonlinear and context-aware interactions.
- **SNR-guided feature aggregation** followed by a learnable nonlinear projection, enhancing robustness against local noise and illumination variation.

We compare different feature fusion strategies in A.5, demonstrating the superiority of our approach.

### A.5.4 Feature Strategy for Global Optimization

To improve the stability of global optimization, the feature selection strategy in **EAG3R** focuses on **Harris corners**, which represent sparse yet highly stable points with strong image gradients. These features provide high-confidence geometric constraints and enhance convergence in the optimization of camera pose and structure. Three strategies are compared: Harris corners (ours), SuperPoint(learned detector), and random sampling.

It is observed in A.6 that random sampling introduces noisy gradients by selecting unreliable regions, thereby degrading optimization stability. In contrast, learned detectors such as SuperPoint are computationally expensive and prone to overfitting when illumination varies significantly. The proposed Harris-based strategy provides a balanced solution, introducing stable and targeted supervision signals that improve convergence while maintaining computational efficiency.

Table A.6: **Comparison of feature selection strategies for global optimization.** Evaluation is conducted on the MVSEC `Night1-3` sequences subsets. Metrics include ATE ↓, RPE trans ↓, and RPE rot ↓. The proposed Harris-corner approach achieves the best trade-off between accuracy and computational efficiency.

| Method | ATE ↓ | RPE trans ↓ | RPE rot ↓ | Computation Cost |
|---|---|---|---|---|
| Random Sampling | 0.687 | 0.261 | 0.153 | Low |
| SuperPoint [12] | 0.685 | 0.260 | 0.153 | High |
| **Harris Corner (Ours)** | **0.655** | **0.236** | **0.152** | Medium |

## A.6  Runtime and Memory Analysis

To assess the computational efficiency of **EAG3R**, we conduct a detailed runtime and memory analysis. Our framework is designed to introduce minimal overhead while maintaining strong reconstruction performance. This efficiency stems from its modular and lightweight architecture components.

**Runtime Analysis.** Table A.7 reports the resource consumption for single image–event pair inference, while Table A.8 summarizes the memory usage during global optimization. Compared to MonST3R, EAG3R introduces only a minor overhead of approximately $+0.4$ GB VRAM, $+0.11$ TFLOPs, and $+1.2$ s per forward pass. Even when incorporating event loss in the global optimization stage, the increase remains modest. These results demonstrate that EAG3R achieves robustness and multimodal integration with minimal computational cost.

Table A.7: Runtime and memory analysis for single image–event pair inference.

| Method | VRAM (GB) | TFLOPs | Forward Time (s) |
|---|---|---|---|
| MonST3R (Baseline) | 2.165 | 1.284 | ∼1.9 |
| +LightUp | 2.192 | 1.287 | ∼2.6 |
| +LightUp + Event Adapter + Fusion (Full) | 2.562 | 1.398 | ∼3.1 |

Table A.8: Memory usage during global optimization (sequence length = 20).

| Method | VRAM (GB) |
|---|---|
| MonST3R (Baseline) | 10.99 |
| EAG3R (w/o event loss) | 12.08 |
| EAG3R (w/ event loss) | 14.02 |

**Scalability to Longer Sequences.** We further analyze scalability with respect to sequence length, following the reviewer's suggestion. EAG3R employs a *sliding-window optimization* scheme, where only pairwise pointmaps and loss terms within a fixed temporal window are computed. This design avoids the quadratic complexity of fully connected graph optimization, maintaining a *constant problem size* per iteration regardless of video length (as is reported in A.9). Consequently, both runtime and memory cost scale linearly with the total number of frames, as confirmed by our experiments running on an NVIDIA A100 GPU.

Table A.9: Scalability analysis: peak memory usage vs. sequence length.

| Sequence Length | 20 | 40 | 60 | 80 | 100 |
|---|---|---|---|---|---|
| **Max Memory (GB)** | 14.02 | 20.19 | 27.78 | 37.40 | 46.49 |

Overall, the results confirm that EAG3R maintains near-linear computational growth with respect to sequence length and introduces only marginal overhead compared to the baseline.

## A.7 Generalization to More Datasets

To demonstrate EAG3R's scalability, we conducted additional experiments on MVSEC indoor and M3ED datasets, covering diverse environments (indoor, outdoor, night, HDR), sensor platforms (drones, robots, cars), and motion types, including complex aerial and ambulatory trajectories.

Models were trained under normal lighting and evaluated in low-light/HDR conditions in a zero-shot setting, confirming EAG3R's robustness beyond vehicle-centric scenes.

**HDR Environments (Robot Dog).** To assess the model's performance in high-dynamic-range (HDR) conditions, we evaluated EAG3R on the challenging *M3ED robot dog* dataset *penno_plaza_lights* split, which features rapid motion and severe illumination fluctuations. As shown in Table A.10, EAG3R achieves substantially higher pose estimation accuracy than both the MonST3R baseline and its scene-finetuned variant across all key metrics.

Table A.10: Pose estimation performance on the M3ED robot dog dataset under HDR lighting conditions.

| Method | ATE ↓ | RPE trans ↓ | RPE rot ↓ |
|---|---|---|---|
| MonST3R | 0.3425 | 0.1919 | 2.3003 |
| MonST3R (finetune) | 0.1853 | 0.0981 | 1.8493 |
| **EAG3R (Ours)** | **0.1361** | **0.0632** | **0.6086** |

**High-Speed Outdoor Drone Scenarios.** To further evaluate robustness under extreme motion and complex lighting, we tested EAG3R on the *M3ED high-speed drone* dataset. As summarized in Table A.11, EAG3R consistently outperforms the baseline in all three outdoor sequences, demonstrating reliable pose estimation even in high-speed and high-contrast conditions.

Table A.11: Pose estimation results on the M3ED high-speed outdoor drone sequences.

| | High Beams | | | Penno Parking 1 | | | Penno Parking 2 | | |
|---|---|---|---|---|---|---|---|---|---|
| Method | ATE ↓ | RPE trans ↓ | RPE rot ↓ | ATE ↓ | RPE (trans) ↓ | RPE (rot) ↓ | ATE ↓ | RPE (trans) ↓ | RPE (rot) ↓ |
| MonST3R | 0.1951 | 0.0668 | 1.0852 | 0.1607 | 0.0942 | 0.4910 | 0.4397 | 0.2502 | **0.9023** |
| **EAG3R (Ours)** | **0.1111** | **0.0572** | **0.6450** | **0.1189** | **0.0748** | **0.5380** | **0.3089** | **0.1572** | 0.9032 |

**High-Speed Indoor Drone Scenarios.** We further evaluated EAG3R's depth estimation performance on the indoor sequences of the MVSEC dataset. As reported in Table A.12, EAG3R achieves the best results across all key depth metrics, surpassing both the baseline and its finetuned variant by a large margin.

## A.8 Statistical Analysis and Robustness Validation

To further assess robustness, we performed statistical analysis across four independent training runs. As summarized in Table A.13, the results exhibit low variance and consistent performance across all key metrics, confirming the model's robustness and stability during optimization.

These results indicate that **EAG3R** maintains consistent performance across multiple random initializations, with all reported metrics exhibiting low variance and statistically significant stability.

## A.9 Limitations

Despite the strong empirical performance of EAG3R, several limitations remain:

**Limited dataset availability.** Currently, there is a scarcity of public datasets that simultaneously provide real event data, RGB videos, and accurate ground-truth geometry. To address this, our future work aims to curate a diverse dataset featuring high-quality, real-world event-RGB pairs across varied lighting and motion scenarios.

**Dependence on event data quality.** Our approach assumes access to temporally aligned, high-quality event streams. Although the SNR-aware fusion mechanism mitigates some effects of noise, the

Table A.12: Depth estimation performance on MVSEC indoor drone sequences.

| Method | Abs Rel ↓ | $\delta < 1.25$ ↑ | RMSE log ↓ |
|---|---|---|---|
| MonST3R | 0.097 | 0.918 | 0.146 |
| MonST3R (finetune) | 0.307 | 0.429 | 0.331 |
| **EAG3R (Ours)** | **0.041** | **0.972** | **0.094** |

Table A.13: **Statistical evaluation of EAG3R across four independent runs.** Reported metrics include absolute relative error (Abs Rel ↓), accuracy under threshold ($\delta < 1.25$ ↑), and log RMSE ↓. Results indicate low standard deviation and statistically significant stability across all sequences.

| Metric | Night1 | | | Night2 | | | Night3 | | |
|---|---|---|---|---|---|---|---|---|---|
| | Abs Rel ↓ | $\delta < 1.25$ ↑ | RMSE log ↓ | Abs Rel ↓ | $\delta < 1.25$ ↑ | RMSE log ↓ | Abs Rel ↓ | $\delta < 1.25$ ↑ | RMSE log ↓ |
| Mean | 0.3546 | 0.4851 | 0.4171 | 0.3137 | 0.4969 | 0.3939 | 0.2904 | 0.5144 | 0.3796 |
| Std | 0.0211 | 0.0059 | 0.0163 | 0.0076 | 0.0340 | 0.0113 | 0.0059 | 0.0321 | 0.0099 |
| $p$-value | 0.0009 | <0.0001 | <0.0001 | <0.0001 | 0.0013 | <0.0001 | <0.0001 | 0.0010 | <0.0001 |

performance of EAG3R can still degrade when event data are excessively sparse, noisy, or misaligned. In particular, we attempted to train our model using synthetic events generated by V2E [20], but observed that the low fidelity of these generated events caused optimization instability, including gradient explosion and failure to converge.

## A.10   Broader Impacts

**Positive impact.**   EAG3R improves 3D perception in challenging environments involving dynamic content and poor illumination. This has the potential to enhance safety and reliability in autonomous systems such as drones, mobile robots, and vehicles, particularly in low-light or fast-motion settings. Additionally, our approach may benefit applications in assistive technology, remote exploration, and AR/VR, where robust scene understanding under non-ideal conditions is critical.

**Potential risks and misuse.**   As with many vision-based systems, there exists a risk of misuse in surveillance or privacy-invasive applications. EAG3R's ability to reconstruct geometry from dark or partially visible scenes could be leveraged in ways that compromise individual privacy. Moreover, due to reliance on event cameras, which remain expensive and less common, the technology may be disproportionately accessible to well-funded institutions, potentially widening gaps in accessibility.

