# OpenReview forum: "EAG3R: Event-Augmented 3D Geometry Estimation for Dynamic and Extreme-Lighting Scenes"
_NeurIPS.cc/2025/Conference — NeurIPS 2025 spotlight_

### Official Review · Reviewer_dFm7 · 2025-06-09

**Clarity:** 3
**Significance:** 2
**Originality:** 1
**Rating:** 4
**Confidence:** 3

**Summary:**

This paper proposes EAG3R, a method for robust 3D geometry estimation (including depth, camera pose, and 4D reconstruction) in challenging scenes with dynamic objects and extreme low-light conditions. The core idea is to augment an existing pointmap-based reconstruction approach, MonST3R, with information from an event camera. The contributions are twofold: 1) A feature fusion module that combines features from RGB images and event streams. 2) A event-based photometric consistency loss that is integrated into the global optimization stage. This loss enforces consistency between the motion predicted by the 3D model and the brightness changes directly observed by the event camera. Experiments on the MVSEC dataset show that EAG3R outperforms RGB-only baselines in zero-shot evaluations on nighttime sequences.

**Questions:**

- The main components of EAG3R appear to be adaptations of existing work (MonST3R backbone, Retinex/SNR from [8, 69], and the event loss model from [21]). Could you please elaborate on what you consider the core conceptual novelty of your work beyond the successful integration of these components? For instance, is there a non-obvious insight or a fundamental challenge in fusing event data with pointmap-based methods?

- The event-based photometric loss is adapted from [21]. Could you discuss the specific challenges encountered when integrating this loss into the global optimization framework of MonST3R? The original work [21] used it for feature tracking. How does its behavior and impact change when used as a global constraint on a dense, globally optimized state (pointmaps, poses, intrinsics)? Were any non-trivial modifications required to make it stable and effective in this new context?

- Your evaluation is limited to the MVSEC dataset, and you mention that training with V2E-generated events failed. This raises questions about the method's generalizability.
    - How do you expect EAG3R to perform on other event-camera datasets, which may have different sensor noise characteristics or scene content?
    - Could you provide more insight into why V2E-generated events caused training to fail?

**Ethical Concerns:**

["NO or VERY MINOR ethics concerns only"]

**Final Justification:**

The paper solves quite a challenging problem of reconstructing 3D scenes in very low light conditions. The rebuttal gave an explanation for the V2E failures. Overall, it's a good systems-level contribution although I would like to have seem more qualitative results of the reconstructions.

**Limitations:**

yes

**Quality:**

3

**Strengths And Weaknesses:**

**Strengths**
- The paper tackles a highly relevant and challenging problem in 3D vision: robust reconstruction in adverse conditions. The proposed method demonstrates significant and consistent quantitative improvements over strong, recent baselines (DUSt3R, MonST3R, Easi3R) on monocular depth and camera pose estimation tasks (Tables 1 and 2). The empirical results are strong, and the ablation study (Table 3) clearly validates the contribution of each component of the proposed system.

- The paper is very well-written and easy to follow. The method is presented logically, and the figures (especially Figs. 2 and 3) provide clear illustrations of the network architecture and the novel loss function. The motivation for using event cameras is well-established, and the overall narrative is compelling.

- The proposed system is technically sound and thoughtfully designed. The idea of using an SNR map to adaptively fuse information from two different sensor modalities is intuitive. Similarly, incorporating a physically-grounded event consistency loss directly into the global optimization step is a clever way to leverage the strengths of event data to regularize the final reconstruction.

**Weaknesses**
- My main concern is that the work, while effective, appears to be a combination of existing techniques rather than proposing a fundamentally new concept. While the integration of these components into a unified framework produces good results, the paper lacks a core contribution in terms of a new algorithm or theoretical insight. I feel it is better suited for a robotics or systems conference like ICRA or RAL. In essence:
    - The backbone (MonST3R) is an existing method.
    - The Retinex-based enhancement and the concept of SNR-aware fusion are inspired by or directly adapted from prior work on low-light enhancement (e.g., [8], [69]).
    - The formulation of the event-based brightness increment model (eq. 9), which is the core of the novel Levent loss, is explicitly stated to follow the generative model from Gehrig et al. [21].

- The experiments are conducted exclusively on the MVSEC dataset. While MVSEC is an appropriate benchmark for this task, relying on a single dataset makes it difficult to assess the true generalizability. The authors mention that they switched from using V2E-generated events due to "gradient explosion," which suggests the method may be sensitive to the specific noise characteristics or data distribution of more complex scenes. Evaluating on other datasets with would have strengthened the paper's claims of robustness.

- Related to the originality concern, the scientific significance is somewhat limited. The paper convincingly shows that event data improves pointmap-based reconstruction in the dark, which is an expected but arguably valuable confirmation. However, it offers less insight into the fundamental principles of multi-modal fusion for this task beyond what was already known. The contribution is more practical and system-oriented.

---

> ### Author Rebuttal · Authors · 2025-07-31
>
> > W1: My main concern is  ...... follow the generative model from Gehrig et al. [21].
> W3: Related to the originality concern, ...... is more practical and system-oriented.
> Q1: The main components  ..... with pointmap-based methods?
>
> Reply: We thank the reviewer for the feedback and would like to clarify the novelty of **EAG3R**. While based on existing frameworks, our work is the **first to integrate asynchronous event streams with the pointmap-based paradigm**, enabling **robust, pose-free 4D reconstruction** in extreme lighting conditions—where prior methods typically fail. This integration demanded resolving several key challenges, which form our core contributions.
>
> ---
>
> ### 1. Solving The Challenge of Extreme-Light Conditions for Geometric Tasks
>
> * **Challenge:** To enable the framework under extreme lighting conditions, a common workaround for low-light scenarios is to apply image enhancement. However, we found this **insufficient for geometric estimation**.
>
> * **Why Enhancement Fails:** Preprocessing with RetinexFormer ("LightUp") yields minimal or negative impact on downstream performance due to artifacts and failure to guide multimodal fusion.
>
> | Method               | Night1 Abs Rel ↓ | δ<1.25 ↑ | RMSE log ↓ | Night2 Abs Rel ↓ | δ<1.25 ↑ | RMSE log ↓ | Night3 Abs Rel ↓ | δ<1.25 ↑ | RMSE log ↓ |
> |----------------------|------------------|-----------|--------------|------------------|-----------|--------------|------------------|-----------|--------------|
> | MonST3R              | 0.370            | 0.373     | 0.497        | 0.309            | 0.469     | 0.409        | 0.317            | 0.453     | 0.418        |
> | MonST3R (light-up)   | 0.370            | 0.369     | 0.503        | 0.316            | 0.451     | 0.431        | 0.329            | 0.441     | 0.444        |
> | **Ours**             | **0.353**        | **0.491** | **0.416**    | **0.307**        | **0.518** | **0.383**    | **0.288**        | **0.533** | **0.371**    |
>
> - **Our Contribution:** We introduce a **Retinex-inspired SNR estimation module**, not just for enhancement but to compute a spatially-varying **confidence map**. To the best of our knowledge, this is the first time a Retinex-based method has been used to generate a confidence map to guide the adaptive fusion of RGB and event data for a 3D geometry task, trusting events more in dark/noisy regions and RGB more where signals are clear.
>
> ---
>
> ### 2. Designing an Efficient and Effective Event Adapter
>
> - **Challenge:** Events are sparse and asynchronous; adapting them effectively without excessive cost or large paired datasets is non-trivial.
>
> - **Why Naive Solutions Fail:** Using a heavy RGB encoder with zero convolution initialization is both inefficient and ineffective.
>
> |              | Abs Rel ↓ | δ<1.25 ↑ | RMSE log ↓ |
> |--------------|------------|-----------|--------------|
> | **Night1**   |            |           |              |
> | Zero_Conv    | 0.377      | 0.446     | 0.449        |
> | **Ours**     | **0.353**  | **0.491** | **0.416**    |
> | **Night2**   |            |           |              |
> | Zero_Conv    | 0.323      | 0.478     | 0.399        |
> | **Ours**     | **0.307**  | **0.518** | **0.383**    |
> | **Night3**   |            |           |              |
> | Zero_Conv    | 0.302      | 0.485     | 0.379        |
> | **Ours**     | **0.288**  | **0.533** | **0.371**    |
>
> - **Our Contribution:** We propose a **lightweight Swin Transformer-based event adapter**, pretrained via self-supervised learning on large-scale event-only datasets. This allows efficient transfer of **motion and edge priors**, enabling fast convergence and strong generalization under data scarcity.
>
> ---
>
> ### 3. Achieving Robust, Principled Feature Fusion
>
> - **Challenge:** RGB and event data differ significantly in density, noise characteristics, and timing. Simple fusion (e.g., concatenation) or nonlinear fusion (attention) is inadequate.
>
> - **Our Contribution:** A **multi-stage adaptive fusion strategy** that:
>   - Uses **cross-attention within the encoder**, where event features query RGB features at multiple levels, learning nonlinear, context-aware relationships.
>   - Combines this with **SNR-guided feature aggregation**, followed by a **learnable nonlinear projection** for further refinement. This ensures robustness to local noise and lighting variation.
>
> | Method               | Night1 Abs Rel ↓ | δ<1.25 ↑ | RMSE log ↓ | Night2 Abs Rel ↓ | δ<1.25 ↑ | RMSE log ↓ | Night3 Abs Rel ↓ | δ<1.25 ↑ | RMSE log ↓ |
> |----------------------|------------------|-----------|--------------|------------------|-----------|--------------|------------------|-----------|--------------|
> | Feature Add          | 0.357            | 0.482     | 0.417        | 0.312            | 0.519     | 0.384        | 0.295            | 0.535     | 0.373        |
> | Last Layer Attention | 0.361            | 0.495     | 0.426        | 0.322            | 0.486     | 0.396        | 0.296            | 0.515     | 0.380        |
> | **Ours**             | **0.353**        | **0.491** | **0.416**    | **0.307**        | **0.518** | **0.383**    | **0.288**        | **0.533** | **0.371**    |
>
> > Q2: The event-based photometric loss ...... make it stable and effective in this new context?
>
> Reply: We thank the reviewer for the insightful comment. We emphasize that adapting event-based photometric consistency loss to a **global optimization framework** is not a trivial integration but a **conceptual and methodological innovation**. Our $L_{event}$ is not a minor extension for local tracking—it is specifically designed for the more complex task of **global 4D reconstruction**.
>
> Our contributions are highlighted by three fundamental differences: **application domain**, **motion modeling**, and **feature point strategy**.
>
> ---
>
> ### 1. Application Domain: From Local 2D Tracking to Global 4D Reconstruction
>
> - Prior work (e.g., Gehrig et al. [21]) focuses on **local 2D feature tracking**, operating on small patches in the image plane.
> - In contrast, EAG3R performs **global 4D scene reconstruction**. Our $L_{event}$ serves as a supervisory loss within a global optimization (similar to bundle adjustment), jointly refining:
>   - **Per-frame depth maps ($D^t$)**
>   - **3D camera poses ($P^t$)**
>   - **Camera intrinsics ($K^t$)**
>
> This moves from localized 2D motion to **global scene-level optimization**.
>
> ---
>
> ### 2. Motion Modeling: From Optical Flow to Geometric Projection
>
> - Previous work models motion via **2D optical flow**, which is unreliable in low-light where RGB quality degrades.
> - Our approach bypasses this by computing the motion field **$\Delta u_{\text{cam}}$** **geometrically** from estimated 3D structure and camera poses:
>
> $$
> \Delta u_{\text{cam}}^{t \rightarrow t'} = \pi(K^{t'} (R^{t \rightarrow t'} (D^t(x) \cdot (K^t)^{-1} \tilde{x}) + T^{t \rightarrow t'})) - x
> $$
>
> - This **physically grounded projection** provides robust supervision even when RGB fails, linking observed events directly to 3D geometry.
>
> ---
>
> ### 3. Feature Strategy: High-Confidence Constraints for Global Optimization
>
> - We focus $L_{event}$ on **Harris corners**—sparse but stable points with strong gradients—introducing **high-quality constraints** into the global optimization.
> - We compared this with:
>   1. **Harris corners** (ours)
>   2. **SuperPoint** (learned detector)
>   3. **Random sampling**
>
> | Method           | ATE ↓ | RPE trans ↓ | RPE rot ↓ | Computation Cost |
> |------------------|:-----:|:-----------:|:---------:|:----------------:|
> | **Harris corner** | **0.655** | **0.236** | **0.152** | Medium            |
> | Random Sampling  | 0.687 | 0.261       | 0.153     | Low               |
> | SuperPoint       | 0.685 | 0.260       | 0.153     | High              |
>
> - Random sampling introduce noisy gradients by selecting unreliable regions, degrading optimization. Network-based detectors are computational expensive. Our Harris-based method ensures **stable, targeted supervision**.
>
> ---
>
> In summary, EAG3R significantly extends previous event-based tracking by redefining the application scope, introducing a geometry-driven motion formulation, and applying a tailored feature strategy—**transforming local principles into a globally optimized 3D reconstruction system**.
>
>
> > W2: The experiments are c ...... f robustness.
> Q3: Your evaluation is ......  caused training to fail?
>
> Reply:  We thank the reviewer for the valuable feedback on dataset diversity.
>
> **Please refer to our reply to our reply to reviewer jzhF W3 for reasons for choosing MVSEC and additional experiments on other datasets.**
>
> #### Why V2E generated datasets fail?
>
> We determined that the failure of V2E) synthesis in low-light conditions stems from two compounding issues within the V2E pipeline: degraded frame interpolation and unstable event triggering.
>  1. Low-Light Frame Interpolation
> The V2E process relies on optical flow-based interpolation to generate high-framerate video. However, low-light footage suffers from high noise and low contrast, which severely degrades the quality of the optical flow estimation. This leads to inaccurate or artifact-ridden interpolated frames, making the synthesized brightness changes unreliable for the subsequent event generation step.
>
> #### 2. Unstable Log-Intensity Event Triggering
> V2E generates events when the change in log-intensity at a pixel crosses a set contrast threshold, $C$:
>
> $$ \Delta L(x, y, t) = \log(I(x, y, t)) - \log(I_{\text{ref}}(x, y)) \geq \pm C $$
>
> In low-light conditions, pixel intensities $I(x, y, t)$ are very low and close to zero. This makes the $\log(I)$ operation numerically unstable and highly sensitive to noise. As a result, the V2E process either fails to generate meaningful events from subtle motion or produces a stream of spurious events triggered by random noise.

---

> > ### Comment · Reviewer_dFm7 · 2025-08-07
> >
> > Thanks for the detailed response. This addresses most of my initial concerns. I will take these into account for my final rating.

---

> > > ### Author Response · Authors · 2025-08-07
> > >
> > > Dear reviewer dFm7,
> > >
> > > We sincerely appreciate your follow-up and are glad that our response has addressed most of your concerns. Should any concerns remain, we would be happy to address them.
> > >
> > > Best regards,
> > >
> > > The Authors of Submission 6067

---

### Official Review · Reviewer_jzhF · 2025-07-01

**Clarity:** 2
**Significance:** 3
**Originality:** 3
**Rating:** 5
**Confidence:** 3

**Summary:**

The paper proposes an evolution of DUSt3R / MonST3R that augments those pipelines with event data. Integrations are done in two primary ways, i.e. via (i) an event perception module and (ii) an event-based photometric consistency loss. The method demonstrates clear performance improvements, particularly in low light scenarios.

**Questions:**

As noted in the previous section, my main concerns revolve around the writing style of the paper. That being said, the work is overall novel and produces clearly good results. I therefore believe it should be accepted, so my preliminary score is borderline accept.

On a separate note, I do wonder if the authors could expand the experiment to include more generic (i.e. non-event-based) datasets, by simulating the event system. For example, it would be interesting to know if a variant of their photometric loss could improve performance when e.g. computed over pairs of RGB images instead of pairs of events and RGB detections.

**Ethical Concerns:**

["NO or VERY MINOR ethics concerns only"]

**Final Justification:**

The rebuttal addressed many of my concerns, so I increased my final review score to Accept.

**Limitations:**

Yes, this is presented in the work.

**Quality:**

3

**Strengths And Weaknesses:**

Positive:
- The contribution is novel -- to my knowledge this is the first work to integrate event data within a dust3r-like formulation.
- The benefits of the event system are fairly well proven by the experimental section. While the tables of quantitative results in the main paper are useful, I found the extra results in the supplementary material section particularly interesting.

Negative:
- I find the paper somewhat difficult to read. For much of the paper, the authors simply describe the architecture and losses, without much intuition. For example, the lightweight event adapter in lines 151-160 effectively describes the figure and architecture, but provide little reason as to why the authors did things the way they did. In my view, more intuition would make the paper easier to read.
- Some of the algorithmic choices are not well well ablated. For example, the authors chose Harris feature to simulate event activations on RGB images. While I understand the overall reason for matching these detections to events, this should be validated experimentally (as there are many other types of features that could work well, or, indeed, the authors could train their own relevant feature detector).
- The experiments have a wide breadth (i.e. across many domains), but are quite shallow (i.e. not much testing within each domain). This is probably explained by the relative low availability of event + RGB datasets.

---

> ### Author Rebuttal · Authors · 2025-07-31
>
> > W1: I find the paper somewhat difficult to read. ...... In my view, more intuition would make the paper easier to read.
>
> Reply: We thank the reviewer for their valuable feedback on improving the readability of our paper. We appreciate the opportunity to clarify our design process, particularly for the event adapter and feature fusion strategy. Here we provide more intuitions and experimental considerations for our architecure and loss design, we will also revise out paper accordingly.
>
> ### 1. Design Rationale for the Hybrid Architecture
>
> Our hybrid architecture design seeks an optimal balance between performance and efficiency. We initially evaluated an approach that duplicated the heavyweight RGB ViT encoder, but experiments showed it incurred significant computational overhead for limited performance gain. As shown in the table below, our final proposed architecture significantly outperforms this initial design.
>
> | | **Night1** | | | **Night2** | | | **Night3** | | |
> | :--- | :---: | :---: | :---: | :---: | :---: | :---: | :---: | :---: | :---: |
> | | **Abs Rel ↓** | **δ<1.25 ↑** | **RMSE log ↓** | **Abs Rel ↓** | **δ<1.25 ↑** | **RMSE log ↓** | **Abs Rel ↓** | **δ<1.25 ↑** | **RMSE log ↓** |
> | **Zero_Conv** | 0.377 | 0.446 | 0.449 | 0.323 | 0.478 | 0.399 | 0.302 | 0.485 | 0.379 |
> | **Ours** | 0.353 | 0.491 | 0.416 | 0.307 | 0.518 | 0.383 | 0.288 | 0.533 | 0.371 |
>
> We ultimately chose a **lightweight Swin Transformer**, pre-trained on large-scale event datasets, as the event backbone. The core advantage of this strategy is its ability to effectively transfer powerful, general-purpose features (e.g., motion patterns) from event data. This facilitates efficient model convergence and stronger generalization, especially in data-scarce, low-light scenarios.
>
> ### 2. Design Rationale for the Event-Based Photometric Loss ($L_{event}$)
>
> Our event-based photometric consistency loss ($L_{event}$) is designed to provide a robust spatiotemporal supervision signal, particularly in scenarios where traditional image-based cues fail due to low light. Its effectiveness and stability stem from two key design choices:
>
> * **Geometrically-Derived Motion**: To circumvent the unreliability of 2D optical flow in low light, the motion field is generated directly via geometric projection from the global 3D state (depth map $D^t$ and camera transformations $P^t, K^t$). This method establishes a direct, physically consistent constraint between the 3D geometry being optimized and the event data.
>
> * **Scale-Invariant Formulation**: To address the unknown and variable contrast sensitivity threshold ($C$) of event cameras, we normalize both observed and predicted brightness-change patches to a unit $L^2$ norm. This makes the optimization objective focus on the **consistency of gradient directions** rather than their absolute magnitudes, ensuring robustness to sensor variations and stability during optimization.
>
>
> > W2: Some of the algorithmic choices ......, the authors could train their own relevant feature detector).
>
> **Reply**: We thank the reviewer for the constructive comment regarding ablation on the choice of Harris corners. We apologize if our rationale was not sufficiently articulated in the submission. Below, we provide both theoretical justification and empirical validation.
>
> **Theoretical Motivation**
>
> Our use of Harris corners is guided by two key considerations:
>
> - **Signal Quality:** Events are triggered by brightness changes, and Harris corners naturally lie in high-gradient regions (e.g., edges, textured areas), which produce dense and reliable event streams. These regions are ideal for enforcing event-based supervision tied to motion dynamics.
>
> - **Numerical Stability:** The event loss in our Levent formulation depends on the image gradient (via brightness constancy). In flat, low-texture areas, gradients approach zero, making the event loss unstable or uninformative. Harris corners ensure we apply the loss only where it is physically meaningful and robust.
>
> While learned detectors like SuperPoint can also identify salient features, Harris offers a simpler, interpretable, and gradient-aligned solution that matches the assumptions behind the event loss.
>
> **Empirical Validation**
>
> To validate this design choice, we performed an ablation study on pose estimation for night1 sequences comparing three patch selection methods:
>
> 1. **Harris corners** (ours)
>
> 2. **SuperPoint corners** (learned detector)
>
> 3. **Random sampling**
>
> Results show that Harris-based selection achieves the best trade-off between performance and stability. SuperPoint yields comparable but slightly lower results with higher computational cost, while random sampling degrades performance due to noisy gradients in textureless regions. This confirms the effectiveness and efficiency of our choice.
>
> | Method | ATE | RPE trans | RPS rot | Computational Cost |
> | :--- | :---: | :---: | :---: | :---: |
> | Harris corner | 0.655 | 0.236 | 0.152 | medium |
> | Random Sampling | 0.687 | 0.261 | 0.153 | low |
> | Superpoint | 0.685 | 0.260 | 0.153 | high |
>
> > W3: The experiments have a wide breadth (i.e. across many domains), but are quite shallow (i.e. not much testing within each domain). This is probably explained by the relative low availability of event + RGB datasets.
>
> Reply: We thank the reviewer for the valuable feedback on dataset diversity. We first provide an overview of mainstream event-RGB datasets and explain our selection rationale. We then present additional experiments on MVSEC Indoor and M3ED, which cover a broad range of environments and motion patterns.
>
> ### The reason for using MVSEC
> Our choice of the MVSEC dataset was guided by the strict requirements of our task: robust 3D geometry estimation in dynamic scenes under extreme lighting. This demands datasets with aligned RGB, event data, and accurate ground truth for both depth and pose — a combination that is rarely available.
> As shown in the table below, few existing datasets satisfy the necessary conditions for evaluating dynamic reconstruction in challenging lighting.
>
> | Dataset | Low-light Scenes | Dynamic Scenes | RGB | Depth Sensor | Ground Truth Pose | Platform | Environment |
> | :--- | :--- | :--- | :--- | :--- | :--- | :--- | :--- |
> | DSEC | ✓ | ✓ | ✓ | LiDAR-16 | X | Car | Outdoor |
> | UZH-FPV | X | ✓ | ✓ | X | MoCap | Drone | Indoor/Outdoor |
> | DAVIS 240C | X | ✓ | ✓ | X | MoCap | Handheld | Indoor/Outdoor |
> | GEN1 | ✓ | ✓ | ✓ | X | X | Car | Outdoor |
> | Prophesee 1MP | ✓ | ✓ | ✓ | X | X | Car | Outdoor |
> | TUM-VIE | ✓ | ✓ | ✓ | X | MoCap | Handheld/Head-mounted | Indoor/Outdoor |
> | MVSEC | ✓ | ✓ | ✓ | LiDAR-16 | MoCap / Cartographer | Car/Drone | Indoor/Outdoor |
> | M3ED | ✓ | ✓ | ✓ | LiDAR-64 | LIO | Car/Legged Robot/Drone | Indoor/Outdoor |
>
> ### Scalability of EAG3R
> To demonstrate EAG3R’s scalability, we conducted additional experiments on MVSEC indoor and M3ED datasets, covering diverse environments (indoor, outdoor, night, HDR), sensor platforms (drones, robots, cars), and motion types, including complex aerial and ambulatory trajectories.
>
> Models were trained under normal lighting and evaluated in low-light/HDR conditions in a zero-shot setting, confirming EAG3R’s robustness beyond vehicle-centric scenes.
>
> #### Performance in HDR environments (robot dog)
>
> To evaluate the model's performance in HDR scenarios, we conducted tests on the demanding pose estimation M3ED robot dog dataset featuring intense lighting changes. As shown in the table below, EAG3R's pose estimation accuracy is generally superior to both the MonST3R baseline and its scene-finetuned version, with substantial improvements across all key metrics.
>
> |  | penno_plaza_lights | | |
> | :--- | :--- | :--- | :--- |
> | | ATE | RPE trans | RPE rot |
> | MonST3R | 0.34253 | 0.19195 | 2.30028 |
> | MonST3r (finetune) | 0.18527 | 0.09806 | 1.84931 |
> | EAG3R | 0.13608 | 0.06317 | 0.60864 |
>
> #### Performance in high-speed outdoor drone scenarios
>
> To further test the model's robustness under high-speed motion and complex lighting, we performed evaluations on M3ED high-speed drone dataset. EAG3R demonstrated robust performance in these highly challenging scenarios. For the demanding pose estimation, EAG3R shows clear improvements over the baseline.
>
> |  | High Beams | | | Penno Parking 1 | | | Penno Parking 2 | | |
> | :--- | :--- | :--- | :--- | :--- | :--- | :--- | :--- | :--- | :--- |
> | | ATE | RPE trans | RPE rot | ATE | RPE trans | RPE rot | ATE | RPE trans | RPE rot |
> | MonST3R | 0.19511 | 0.06677 | 1.08519 | 0.16074 | 0.09418 | 0.491 | 0.43969 | 0.25023 | 0.90225 |
> | EAG3R (ours) | 0.11108 | 0.05717 | 0.64498 | 0.1189 | 0.07483 | 0.53798 | 0.30893 | 0.15723 | 0.90318 |
>
> #### Performance in high-speed indoor drone scenarios
>
> We also evaluated the depth estimation performance on the indoor sequence from the MVSEC dataset. As shown in the table below, our proposed method, EAG3R, achieves the best performance across all key metrics compared to the MonST3R baseline and its finetuned variant.
>
> | Method | Abs Rel ↓ | δ<1.25 ↑ | RMSE log ↓ |
> | :--- | :---: | :---: | :---: |
> | MonST3R | 0.097 | 0.918 | 0.146 |
> | MonST3R (finetune) | 0.307 | 0.429 | 0.331 |
> | EAG3R (Ours) | 0.041 | 0.972 | 0.094 |

---

> > ### Comment · Reviewer_jzhF · 2025-08-05
> >
> > Thank you for the updated notes in the rebuttal. These addresses many of my concerns, so I am raising my score to 5.

---

> > > ### Author Response · Authors · 2025-08-05
> > >
> > > Dear Reviewer jzhF,
> > >
> > > We are delighted that our responses and experiments addressed your questions and concerns, and we sincerely thank you for raising your score!
> > >
> > > Best regards,
> > >
> > > The Authors of Submission 6067

---

### Official Review · Reviewer_b25i · 2025-07-01

**Clarity:** 4
**Significance:** 4
**Originality:** 4
**Rating:** 4
**Confidence:** 4

**Summary:**

This paper presents EAG3R, an event-augmented framework for robust 3D geometry estimation in dynamic and extreme-lighting scenes. Leveraging asynchronous event streams, EAG3R enhances the MonST3R backbone with two key innovations: a Retinex-inspired image enhancement module combined with a lightweight event adapter for SNR-aware feature fusion, and an event-based photometric consistency loss to enforce spatiotemporal coherence during global optimization. This approach enables effective integration of event and RGB features, adapting to local reliability to maintain performance in low-light conditions without retraining for nighttime scenarios. Experiments on the MVSEC dataset demonstrate that EAG3R outperforms state-of-the-art RGB-only baselines in monocular depth estimation, camera pose tracking, and dynamic reconstruction, achieving up to 10.7% lower Absolute Relative Error in extreme low-light scenes and showcasing strong zero-shot generalization to nighttime environments.

**Questions:**

1. The method relies on Harris corners for event patches, which may miss structural details in texture-sparse regions (e.g., walls, smooth surfaces). This could lead to incomplete reconstructions—validate with alternative patch selection (e.g., random sampling) to quantify bias.
2. Experiments only use MVSEC,  more validation is necessary.
3. The paper claims dynamic scene support but doesn’t quantify errors for moving objects (e.g., vehicles).
4. The SNR-aware fusion assumes linear weightings, but lighting nonlinearities (e.g., saturation) may invalidate this. Introduce nonlinear fusion models (e.g., attention-based) and compare via ablation.
5. Ignoring negative events (only using positive polarity) might lose intensity decrease signals in sudden lighting changes.

**Ethical Concerns:**

["NO or VERY MINOR ethics concerns only"]

**Limitations:**

Please refer to the Weaknesses and Limitations parts.

**Paper Formatting Concerns:**

The reference formatting is non-standard. For instance, some journal names are presented in full, while others use abbreviations.

**Quality:**

4

**Strengths And Weaknesses:**

Strengths
EAG3R effectively integrates event streams with RGB data, using a Retinex-inspired module to enhance low-light images and an SNR-aware fusion strategy to adaptively combine features. This allows robust 3D geometry estimation in dynamic, extreme-lighting scenes without retraining for nighttime conditions. The event-based photometric consistency loss improves spatiotemporal coherence, and experiments on MVSEC show significant performance gains over RGB-only baselines in depth estimation, pose tracking, and dynamic reconstruction.
Weaknesses
The method relies on Harris corners for event patch selection, which may miss structural details in texture-sparse areas. The framework does not explicitly address reconstruction errors for moving objects, leaving room for improvement in handling dynamic scenes.

---

> ### Author Rebuttal · Authors · 2025-07-31
>
> > Q1: The method relies on Harris corners...... to quantify bias.
>
> Reply: We thank the reviewer for the valuable comment. While Harris corners may overlook texture-sparse regions, our design deliberately uses event loss as a **sparse, high-confidence supplement**, not the primary geometric constraint. These regions are already addressed by our global optimization, which integrates alignment, smoothness, and optical flow losses to ensure spatial-temporal consistency.
>
> 1. Why choose Harris corners
> Under extreme lighting where optical flow supervision becomes unreliable (due to degraded RAFT performance), the event loss becomes crucial. However, applying it indiscriminately in low-gradient regions like walls can lead to unstable or uninformative gradients. Hence, we restrict it to Harris corners—locations with strong gradients and dense, reliable event activity—maximizing both **signal quality** and **numerical stability**.
>
> 2. Comparing Harris corner-based path selection with other feature selection approaches
> To assess potential bias, we conducted an ablation comparing Harris-based and random patch sampling. As is summarized in the following Table, random sampling caused noticeable degradation, as many sampled patches fell into low-texture regions, introducing noisy gradients that hindered convergence. This empirically confirms that our targeted strategy enhances effectiveness while avoiding spurious influence from weak or invalid supervision.
>
> | Method | ATE | RPE trans | RPS rot |
> | :--- | :---: | :---: | :---: |
> | Harris corner | 0.655 | 0.236 | 0.152 |
> | Random Sampling | 0.687 | 0.261 | 0.153 |
> | Superpoint | 0.685 | 0.260 | 0.153 |
>
> > Q2: Experiments only use MVSEC, more validation is necessary.
>
> **Reply**: Please refer to our reply to reviewer jzhF W3.
>
> > Q3: The paper claims dynamic scene support but doesn’t quantify errors for moving objects (e.g., vehicles).
>
> Reply: We thank the reviewer for the insightful question regarding evaluation on moving objects. The main challenge lies in the lack of dense annotations (e.g., masks) for dynamic objects in current datasets, which makes direct evaluation difficult. Nonetheless, we provide several indirect but effective metrics in the paper to assess performance in dynamic scenes.
>
> * On one hand, in dynamic environments, the accuracy of monocular pose estimation serves as a strong indirect quantitative indicator of the model's ability to disentangle ego-motion from object motion. Poor handling of moving objects would result in incorrect camera motion estimates. As shown in Table 2, our model significantly outperforms the baseline in low-light dynamic settings.
> * On the other hand, we quantify the video depth metric in Table A1 of the Appendix (below), which evaluates the depth estimation accuracy across all pixels in the video sequence. This naturally includes dynamic objects in the scene, thus indirectly reflecting the model’s performance on moving objects.
>
> | Method | Night 1 Abs Rel ↓ | δ<1.25 ↑ | Log RMSE ↓ | Night 2 Abs Rel ↓ | δ<1.25 ↑ | Log RMSE ↓ | Night 3 Abs Rel ↓ | δ<1.25 ↑ | Log RMSE ↓ |
> | :--- | :--- | :--- | :--- | :--- | :--- | :--- | :--- | :--- | :--- |
> | DUST3R | 0.432 | 0.374 | 0.547 | 0.410 | 0.397 | 0.493 | 0.510 | 0.322 | 0.554 |
> | MonST3R | 0.380 | 0.388 | 0.486 | 0.299 | 0.494 | 0.388 | 0.296 | 0.499 | 0.392 |
> | Easi3Rdust3r | 0.427 | 0.388 | 0.549 | 0.435 | 0.376 | 0.515 | 0.504 | 0.324 | 0.566 |
> | Easi3Rmonst3r | 0.375 | 0.381 | 0.484 | 0.308 | 0.490 | 0.397 | 0.314 | 0.465 | 0.404 |
> | EAG3R (Ours) | 0.372 | 0.443 | 0.480 | 0.304 | 0.514 | 0.403 | 0.281 | 0.542 | 0.390 |
>
>
> > Q4: The SNR-aware fusion assumes linear weightings, but lighting nonlinearities (e.g., saturation) may invalidate this. Introduce nonlinear fusion models (e.g., attention-based) and compare via ablation.
>
> Reply: We thank the reviewer for this insightful comment regarding our feature fusion strategy. We would like to clarify that our framework does not limit fusion to a single linear step. Instead, we adopt multi-stage, nonlinear and adaptive mechanisms to robustly integrate RGB and event features:
>
> * Nonlinear fusion in the encoder: We embed a multi-level cross-attention mechanism within the encoder, where event features query RGB features across resolutions. This enables content-aware, nonlinear fusion early in the network, allowing dynamic emphasis on salient information from both modalities.
> * Learnable nonlinear projection after SNR-guided aggregation: The initial SNR-based weighting is followed by a learnable nonlinear projection applied to concatenated features. This step adaptively reweights and transforms the representation, helping address challenges like local lighting nonlinearity.
>
> To directly address the reviewer's suggestion, we conducted new ablation experiments on depth estimation comparing different fusion strategies: linear addition, attention-based fusion at the final stage, and our proposed multi-stage fusion. The results show that our approach outperforms all other choices.
>
> | Methods | night1 Abs Rel ↓ | δ<1.25 ↑ | RMSE log ↓ | night2 Abs Rel ↓ | δ<1.25 ↑ | RMSE log ↓ | night3 Abs Rel ↓ | δ<1.25 ↑ | RMSE log ↓ |
> | :--- | :--- | :--- | :--- | :--- | :--- | :--- | :--- | :--- | :--- |
> | Feature Add | 0.357 | 0.482 | 0.417 | 0.312 | 0.519 | 0.384 | 0.295 | 0.535 | 0.373 |
> | Last Layer Attention | 0.361 | 0.495 | 0.426 | 0.322 | 0.486 | 0.396 | 0.296 | 0.515 | 0.380 |
> | Ours | 0.353 | 0.491 | 0.416 | 0.307 | 0.518 | 0.383 | 0.288 | 0.533 | 0.371 |
>
> > Q5: Ignoring negative events (only using positive polarity) might lose intensity decrease signals in sudden lighting changes.
>
> Reply: We appreciate the reviewer’s concern. However, we would like to clarify that our method does not ignore negative events. Both positive and negative polarity events are used throughout our pipeline.
> To avoid any further confusion, we will revise the manuscript to clearly state that both polarities of events are utilized during feature extraction and fusion.
>
> > Paper Formatting Concerns: The reference formatting is non-standard. For instance, some journal names are presented in full, while others use abbreviations.
>
> Reply: We thank the reviewer for your thorough review and valuable feedback. We sincerely apologize for the inconsistencies in the citation formatting. In the revised version of the paper, we will carefully review and standardize all references to ensure consistency.

---

### Official Review · Reviewer_DUi3 · 2025-07-02

**Clarity:** 3
**Significance:** 3
**Originality:** 3
**Rating:** 5
**Confidence:** 2

**Summary:**

This paper introduces EAG3R, a novel 3D geometry estimation framework that augments pose-free pointmap-based reconstruction with asynchronous event streams. Built on the MonST3R backbone, EAG3R integrates a Retinex-inspired image enhancement module, a lightweight Swin Transformer-based event adapter with SNR-aware fusion, and an event-based photometric consistency loss into the 3D geometry optimization pipeline.

**Questions:**

1. The method assumes perfectly time-synchronized, high-SNR event data aligned with RGB frames. But in practice event cameras can introduce more sensor noise relative to conventional cameras. Have you studied the effect of event noise on your method?
1. Do you have a runtime analysis and GPU memory usage comparison against MonST3R to assess practical feasibility?
1. Can you include a qualitative failure-case analysis or limitations section focused on known challenges with event data?
1. Could you provide examples of situations where EAG3R fails or performs poorly? For example, does it degrade in highly deformable motion, complete darkness with no RGB signal, or heavy event noise scenarios? A qualitative analysis would clarify boundaries of robustness.
1. How does EAG3R handle regions with few or no events, such as textureless walls or static background areas in dynamic scenes? Does the SNR-aware fusion default to purely RGB features in such regions, and if so, is there a risk of low-quality depth in underexposed areas?

**Ethical Concerns:**

["NO or VERY MINOR ethics concerns only"]

**Final Justification:**

The authors have clearly and thoroughly addressed the raised concerns with strong empirical evidence and detailed analysis. EAG3R shows robust performance under noisy and sparse event conditions, as well as strong generalization across datasets and motion patterns. While the method introduces some non-negligible computational overhead, the authors provide runtime and scalability analyses that support its practical feasibility. Statistical validation from multiple training runs confirms that the results are stable and reproducible. The adaptive fusion mechanism effectively handles regions with few or no events, and the paper includes a transparent discussion of limitations and failure cases. Overall, this is a well-executed and meaningful contribution to event-augmented 3D reconstruction.

**Limitations:**

Further study into the following would illuminate potential limitations of the method:
1. Analysis on out-of-distribution motion patterns: MVSEC mostly covers automotive driving scenarios with relatively predictable ego-motion and scene dynamics. It’s unclear how the method handles motion outside of this range.
1. Reliance on event data quality and the method's failure modes: analysis of the method's dependence on event data quality, sensor properties, and failure cases in event-sparse or highly noisy scenarios would be valuable.
1. Scalability to longer sequences/larger scenes: the method performs global optimization over per-frame pointmaps, camera poses, and intrinsics, which is computationally intensive. While the paper reports 300 optimization iterations, it doesn’t quantify how this scales with longer sequences.

**Quality:**

3

**Strengths And Weaknesses:**

## Strengths
1. Relevant problem: 3D geometry estimation under dynamic, extreme low-light conditions is an important challenge for applications like autonomous navigation and AR/VR.
1. Novel technical contributions: retinex-inspired image enhancement module, SNR-aware fusion mechanism between event-based and image-based features and event-based photometric consistency loss incorporated into global optimization.
1. Clear integration of event streams into a contemporary pointmap-based framework (MonST3R), a meaningful step for practical scene reconstruction.

## Weaknesses
1. Limited dataset diversity: all experiments are conducted only on MVSEC outdoor sequences. While the paper justifies this, testing on additional diverse benchmarks (like E3D or open event datasets) would strengthen generalizability claims.
1. While the method’s zero-shot performance on different lighting conditions is a strength, its generalization to other types of scene dynamics (e.g., different motion types, varied object categories, or other outdoor/indoor environments) isn’t demonstrated. A limitation if claiming "general-purpose dynamic reconstruction".
1. Ablation completeness: while ablation studies are helpful, including runtime analysis (added compute/memory overhead from events, fusion, optimization) would clarify practical trade-offs.
1. Event data limitations: the method's performance in highly noisy or sparse event streams (common in commercial-grade sensors) isn’t deeply evaluated.
1. Although the paper claims statistical rigor in the checklist, the reported tables (Tables 1–3) don't visibly include error bars, confidence intervals, or p-values.

---

> ### Author Rebuttal · Authors · 2025-07-31
>
> > W1: Limited dataset diversity
> W2: General-purpose dynamic reconstruction
> L1: Analysis on out-of-distribution motion patterns
>
> Please refer to our reply to jzhF W3.
>
> > W3: Ablation completeness: ... clarify practical trade-offs.
> Q2: Do you have a runtime analysis ... assess practical feasibility?
>
> **Reply:** We thank the reviewer for suggesting runtime analysis.
>
> 1. **Lightweight event fusion**
>   - **Event Adapter**: Uses efficient Swin Transformer, freezing the image encoder and updating only the adapter/decoder, minimizing computation.
>   - **Enhancement Module**: Shallow, Retinex-inspired design ensures high efficiency.
>
> 2. **Runtime analysis**
>   - **Table 1**: For single image-event pairs, EAG3R adds ~0.4 GB VRAM, ~0.11 TFLOPs, and ~1.2s inference time over MonST3R, with potential CUDA optimization.
>   - **Table 2**: For global optimization (sequence length 20), event-guided approach adds ~1 GB VRAM overhead compared to baseline without event loss.
>
>
> Our design introduces minimal computational overhead while maintaining efficiency.
>
> | Method | VRAM Usage (GB) | TFLOPS | Forward Pass Time |
> | :--- | :--- | :--- | :--- |
> | MonST3R (Baseline) | 2.165 | 1.284 | ~1.9s |
> | +Lightup | 2.192 | 1.287 | ~2.6s |
> | +Light up + Event Adapter + Fusion (Full) | 2.562 | 1.398 | ~3.1s |
>
> | Method | VRAM Usage (GB) |
> | :--- | :--- |
> | MonST3R (Bseline) | 10.99 |
> | EAG3R (w/o event loss) | 12.08 |
> | EAG3R (w/ event loss) | 14.02 |
>
> > W4: Event data limitations:...... isn’t deeply evaluated.
> Q1: The method assumes perfectly time-synchronized, ...... the effect of event noise on your method?
> L2: Reliance on event data quality ...... noisy scenarios would be valuable.
>
> **Reply:** We thank the reviewer for their comment on event data quality challenges. Our response covers two aspects:
>
> 1. **MVSEC Noise**: MVSEC’s real-world data, collected with a commercial DAVIS346 sensor, includes inherent noise and ~90% sparsity. EAG3R’s strong performance on this noisy input demonstrates its robustness.
>
> 2. **Noise and Sparsity Simulations**:
> 	- **Noise**: We injected 5%, 10%, and 15% Poisson noise into event streams to mimic sensor noise, testing EAG3R’s performance.
> 	- **Sparsity**: We simulated low-contrast scenes by randomly dropping 5%, 10%, and 15% of events to evaluate robustness.
>
> These tests confirm EAG3R’s resilience to practical noise and sparsity challenges.
>
> | | | **night1** | | | **night2** | | | **night3** | | |
> | :--- | :--- | :--- | :--- | :--- | :--- | :--- | :--- | :--- | :--- | :--- |
> | **Type** | **level** | **Abs Rel** | **δ<1.25** | **RMSE log** | **Abs Rel** | **δ<1.25** | **RMSE log** | **Abs Rel** | **δ<1.25** | **RMSE log** |
> | | **Real** | 0.3526 | 0.4906 | 0.4160 | 0.3066 | 0.5181 | 0.3829 | 0.2875 | 0.5336 | 0.3710 |
> | **Noise** | **+5%** | 0.3526 | 0.4909 | 0.4161 | 0.3066 | 0.5184 | 0.3830 | 0.2876 | 0.5336 | 0.3711 |
> | **Noise** | **10%** | 0.3524 | 0.4923 | 0.4162 | 0.3066 | 0.5188 | 0.3830 | 0.2878 | 0.5338 | 0.3713 |
> | **Noise** | **15%** | 0.3521 | 0.4951 | 0.4164 | 0.3067 | 0.519 | 0.3832 | 0.2881 | 0.5339 | 0.3715 |
> | **Sparsity** | **5%** | 0.3527 | 0.4896 | 0.4161 | 0.3066 | 0.5167 | 0.3831 | 0.2874 | 0.5323 | 0.3710 |
> | **Sparsity** | **10%** | 0.3527 | 0.4883 | 0.4161 | 0.3067 | 0.5150 | 0.3833 | 0.2872 | 0.5310 | 0.3710 |
> | **Sparsity** | **15%** | 0.3527 | 0.4869 | 0.4160 | 0.3067 | 0.5130 | 0.3835 | 0.2871 | 0.5293 | 0.3709 |
>
> Modern event cameras (e.g., Prophesee GenX320) have higher dynamic range and lower noise than the DAVIS346 used in MVSEC, suggesting EAG3R’s performance could improve further with newer hardware.
>
> > W5: Although the paper claims ...... confidence intervals, or p-values.
>
> **Reply:** We thank the reviewer for suggesting statistical analysis. While large-scale 3D vision models like ours, DUSt3R, and MonST3R typically report single-run results due to high computational costs, we conducted 4 additional independent training runs from random initialization to validate robustness. Results show stable performance in depth estimation, with low standard deviation across key metrics.
>
> | | **night1** | | | **night2** | | | **night3** | | |
> | :--- | :--- | :--- | :--- | :--- | :--- | :--- | :--- | :--- | :--- |
> | | **Abs Rel** | **δ<1.25** | **RMSE log** | **Abs Rel** | **δ<1.25** | **RMSE log** | **Abs Rel** | **δ<1.25** | **RMSE log** |
> | **Mean** | 0.3546 | 0.4851 | 0.4171 | 0.3137 | 0.4969 | 0.3939 | 0.2904 | 0.5144 | 0.3796 |
> | **Std** | 0.0211 | 0.0059 | 0.0163 | 0.0076 | 0.034 | 0.0113 | 0.0059 | 0.0321 | 0.0099 |
> | **p-value**| 0.0009 | < 0.0001 | < 0.0001 | < 0.0001 | 0.0013 | < 0.0001 | < 0.0001 | 0.001 | < 0.0001 |
>
>
> > Q3: Can you include a qualitative failure ...... with event data?
> Q4: Could you provide examples of situations ...... boundaries of robustness.
>
> **Reply**:  We thank the reviewer for their insightful comments on our method’s operational boundaries. Our analysis of EAG3R’s performance on challenging MVSEC and M3ED datasets clarifies its robustness:
> 1. **MVSEC Performance**: In nighttime sequences, large motion (e.g., sharp turns) and sparse co-located RGB/event signals (e.g., dark, static areas) challenge performance. However, EAG3R’s degradation is less severe than MonST3R, showing the advantage of event-augmentation.
>
> 2. **M3ED Generalization**: Tests on diverse indoor and M3ED datasets (e.g., drone and quadruped robot motion) confirm EAG3R’s robustness across complex trajectories, outperforming baselines and avoiding overfit to MVSEC’s vehicular patterns.
>
> 3. **Failure Cases**: Extreme conditions like complete darkness or heavy event noise may cause failures due to limited information.
>
> In summary, while aggressive motion and sparse events pose challenges, EAG3R consistently outperforms pure visual methods and generalizes well across diverse real-world scenarios.
>
> > Q5: How does EAG3R handle regions with few or no events, such as textureless walls or static background areas in dynamic scenes? Does the SNR-aware fusion default to purely RGB features in such regions, and if so, is there a risk of low-quality depth in underexposed areas?
>
> **Reply**: Reply: We thank the reviewer for their insightful comment on our fusion design’s performance. EAG3R’s multi-stage fusion robustly handles event-sparse regions by:
>
> - **Well-lit static areas**: Few events but high-quality RGB input; SNR-aware fusion prioritizes reliable image features.
>
> - **Underexposed static regions**: Low event and RGB quality; early-stage cross-attention fusion uses broader image context to enhance weak local patches.
>
> - **Extreme cases**(e.g., night sky): Unreliable depth estimates are masked out in benchmarks like MVSEC, minimally impacting metrics.
>
> EAG3R’s fusion, combining early cross-modal interaction and SNR-aware aggregation, ensures resilience in low-activity regions, avoiding catastrophic failure under challenging lighting.
>
> > L3: Scalability to longer sequences/larger scenes: the method performs global optimization over per-frame pointmaps, camera poses, and intrinsics, which is computationally intensive. While the paper reports 300 optimization iterations, it doesn’t quantify how this scales with longer sequences.
>
> Reply: We appreciate the reviewer’s comment on scalability. Our EAG3R framework ensures scalability for long video sequences using a sliding window optimization strategy, like MonST3R, computing pairwise pointmaps and loss terms only within the window, avoiding the costly fully connected graph of all frame pairs.
>
> * Constant Problem Size: The sliding window ensures that the optimization problem's size and complexity remain constant at each step, regardless of the total length of the video sequence.
> * Decoupled Iteration Cost: Because the optimization problem is fixed in size, the number of iterations required for convergence does not need to increase for longer videos. The cost per iteration is effectively decoupled from the total sequence length.
>
> The computational cost for processing a long video scales linearly with frame count, not quadratically, ensuring high scalability. Supplementary experiments on an A100 GPU confirm this linear relationship between sequence length and cost.
>
> | Sequence length | Max Memory (GB) |
> | :--- | :--- |
> | 20 | 14.02 |
> | 40 | 20.19 |
> | 60 | 27.78 |
> | 80 | 37.40 |
> | 100 | 46.49 |

---

> > ### Comment · Reviewer_DUi3 · 2025-08-06
> >
> > Dear Authors, thank you for your detailed responses; they have been most helpful. The majority of my concerns have been addressed and so I will raise my score to a 5.

---

> > > ### Author Response · Authors · 2025-08-06
> > >
> > > Dear Reviewer DUi3,
> > >
> > > We are delighted that our responses and experiments addressed your questions and concerns, and we sincerely thank you for raising your score!
> > >
> > > Best regards,
> > >
> > > The Authors of Submission 6067

---

### Note · Authors · 2025-08-13

We sincerely thank the area chair for managing the review process and all reviewers for their constructive feedback. We are encouraged by all reviewers’ support for accepting our paper. Below, we summarize each reviewer’s concerns, our rebuttals, and the post-discussion feedbacks.

|**Reviewer**|**Concerns**|**Our Rebuttal**|**Feedback**|
|:-|:-|:-|:-|
|**DUi3**|**A.** impact of event noise **B.** generalization beyond MVSEC **C.** runtime/memory overhead **D.** robustness in extreme regions **E.** statistical significance **F.** scalability to longer sequences|**A.** noise/sparsity tests demonstrating strong robustness **B.** expanded evaluations on M3ED and MVSEC indoor datasets **C.** runtime/memory analysis showing minimal overhead **D.** qualitative analysis of failure case **E.** multi-run stats with p-values **F.** linear scalability via sliding-window optimization|Concerns addressed; raise score to 5|
|**b25i**|**A.** Harris corner for event patch selection **B.** generalization beyond MVSEC **C.** fusion strategy clarity **D.** dynamic object reconstruction accuracy **E.** polarity usage|**A.** ablations show our method outperforms other alternatives **B.**  expanded evaluations on M3ED and MVSEC indoor datasets **C.** fusion ablations confirm multi-stage method's superiority **D.** video depth metrics for dynamics **E.** both polarities used|Supports acceptance|
|**jzhF**|**A.** design intuition clarity **B.** validation of the Harris corner selection strategy **C.** the wide breadth yet limited depth of evaluation due to dataset availability|**A.** detailed design rationale with ablations **B.** theoretical and empirical analysis for Harris vs learned alternative comparison **C.** expanded evaluation on MVSEC indoor and M3ED|Concerns addressed; raise score to 5|
|**dFm7**|**A.** design rationale **B.** novelty of photometric-consistency loss **C.** generalization beyond MVSEC and v2e failure analysis|**A.** clarified key challenges and detailed our contribution with ablations **B.** novelty explained from domain, motion, and feature perspectives with ablations **C.** explained dataset scarcity, extended evaluations on M3ED and MVSEC indoor datasets, and failure analysis|Concerns addressed with no further questions; consider raising final rating|

We sincerely thank the area chair and all reviewers again for their invaluable feedback and effort.

---

### Decision · Program_Chairs · 2025-09-17

**Decision:**

Accept (spotlight)

**Comment:**

After rebuttal, the reviewers were satisfied with the responses from the authors and upgraded their ratings.
Finally all the reviewers agreed to acceptance.
The authors are encouraged to reflect the responses into the camera ready version.